# Development of Oral Microemulsion Spray Containing Pentacyclic Triterpenes-Rich *Centella asiatica* (L.) Urb. Extract for Healing Mouth Ulcers

**DOI:** 10.3390/pharmaceutics14112531

**Published:** 2022-11-20

**Authors:** Vilasinee Sanguansajapong, Pajaree Sakdiset, Panupong Puttarak

**Affiliations:** 1Department of Pharmacognosy and Pharmaceutical Botany, Faculty of Pharmaceutical Sciences, Prince of Songkla University, Songkhla 90110, Thailand; 2Drug and Cosmetics Excellence Center, Walailak University, Nakhon Si Thammarat 80161, Thailand; 3School of Pharmacy, Walailak University, Nakhon Si Thammarat 80161, Thailand; 4Phytomedicine and Pharmaceutical Biotechnology Excellence Center, Faculty of Pharmaceutical Sciences, Prince of Songkla University, Songkhla 90112, Thailand

**Keywords:** *Centella asiatica* (L.) Urb., fibroblast, oral spray, microemulsion, oral ulcers

## Abstract

Several publications have shown that *Centella asiatica* (L.) Urb. and its active constituents (pentacyclic triterpenes) are effective in wound healing. The pentacyclic triterpenes-rich *C. asiatica* extract (PRE) was prepared following a previous study by microwave-assisted extraction (MAE) and fractionation with macroporous resin. This method provided the pentacyclic triterpene content in the extract up to 59.60% *w*/*w*. The PRE showed potent anti-inflammatory activity by inhibiting nitric oxide (NO) production with an IC_50_ value of 20.59 ± 3.48 μg/mL and a potent fibroblast proliferative effect (165.67%) at concentrations of 10 μg/mL. The prepared microemulsion consisted of a water: oil: surfactant mixture of 2: 2: 6, using coconut oil: clove oil (1:1) as the oil phase and Tween 20: Span 20 (2:1) as the surfactant mixture and 1.0, 2.5, and 5.0% PRE. Cell proliferation, migration, and collagen production of the microemulsion base and microemulsions containing 1.0%, 2.5%, and 5.0% PRE were evaluated. The results revealed that the microemulsion containing 1% PRE had the highest proliferation effect (136.30 ± 3.93% to 152.65 ± 3.48% at concentrations of 10 μg/mL), migration activities (100.00 ± 0.0% at 24 h), and collagen production in human dermal fibroblast (HDF) and human gingival fibroblast (HGF) cells when compared with other formulations or blank. Moreover, the anti-inflammatory activity of microemulsions containing 1% PRE was slightly lower than standard indomethacin. Anti-inflammation of the microemulsion containing PRE exhibited a dose-dependent trend, while 5% PRE was more potent than the standard drug. Considering the potent wound-healing activities and the good anti-inflammatory activity of the microemulsion containing PRE, the microemulsion with 1% PRE was identified as the most suitable oral spray formulation for oral ulcer treatment.

## 1. Introduction

Oral ulcers are breaks in the continuity of the oral epithelium brought about by molecular necrosis [1]. They can result from several causes, such as recurrent aphthous stomatitis, physical trauma, chemical trauma, acute necrotizing ulcerative gingivostomatitis, drug reaction, infection, haematinic deficiency, and immune system disorders [2]. The global prevalence of oral ulcers is about 4% [3]. Patients treated with radiotherapy suffered from oral ulcers at around 80%. Patients treated with chemotherapy were found to experience oral ulcers at 40–80%, and over 75% of those who had bone marrow transplantation were found to suffer from oral ulcers [4]. An oral ulcer causes the patient to suffer from severe pain in the oral cavity. This discomfort is considered one of the adverse effects that may severely impact patient quality of life [2].

Nowadays, a wide range of topical preparations are promoted as oral ulcer treatments. While some patients may find these products helpful, there is little scientific evidence to support their claimed efficacy [2]. Various studies have shown positive properties of the products against mucositis, including local anesthetic, wound healing, antioxidant, anti-microbial, and anti-inflammatory effects [4]. There are many interventions to manage oral mucositis, but there are no accepted standard interventions as the conditions develop under various factors [5]. Some drugs are costly, and the chemical agents may be harmful. Herbal products are popular since they are generally considered to be acceptably safe and to be a lower risk than their synthetic chemical counterparts. Herbal product use can also help reduce the overuse of medications and synthetic chemicals. Many medicinal plants exhibit a wide range of pharmacological activities related to oral ulcer treatment. These herbal raw materials can be found in Thailand, contributing to lower prices. The outstanding features of herbs are efficiency and safety because of their long ethnopharmacological history.

Several herbal medicines have been investigated and demonstrated to be active ingredients in the treatment of oral ulcers, such as *Aloe vera* (L.), *Capsicum annuum* (L.), *Carica papaya* (L.), *Curcuma longa* (L.), and *Matricaria chamomilla* (L.) [6]. In Thailand, *Centella asiatica* (L.) Urb. has been introduced in primary health care to treat burns and scalds, according to the 6th Public Health Development Plan [7]. In the Thailand National List of Essential Medicines (NLEM), *C. asiatica* is recommended as an oral medicine to treat fever and oral ulcers as well as a topical medication to be applied to the skin for wound healing [8]. Most of the *C. asiatica* plant is utilized, including the stems, leaves, and aerial parts. *C. asiatica* is also used in Ayurveda or Traditional Indian Medicine for treating the central nervous system, skin, and gastrointestinal diseases [9].

A number of publications have revealed the effectiveness and functional properties of *C. asiatica* and its active constituents, most notably pentacyclic triterpenes (asiaticoside, madecassoside, asiatic acid, and madecassic acid). These include promotion of the growth of fibroblast cells and collagen production [10]; wound-healing [11], anti-inflammation [12], antibacterial [13], and antioxidant properties [14]; immune system stimulation [15]; neuroprotective activity [16]; hepatoprotective [17] and anti-tumor properties [18].

The medication properties of drugs for curing oral ulcers include wound-healing, anti-inflammation, antioxidation, and antimicrobial activity. These properties were also found in standardized *C. asiatica* extracts and their active constituents [19]. Therefore, this research used standardized *C. asiatica* extracts as the active substance in the formulation of mouth spray. Standardized *C. asiatica* extracts have the potential to be developed as products to prevent and heal mouth ulcers or oral mucosal inflammation efficiently. However, some standardized extracts demonstrated low solubility, low absorption rate, and unfavorable smell and taste [19,20]. According to our preliminary study, a limited amount of the pentacyclic triterpenes-rich *C. asiatica* extract (PRE) (0.1% *w*/*w*) could be incorporated into the aqueous solution. Higher concentrations of the extract incorporated caused precipitation of the undissolved moieties. The PRE consisted of asiatic acid and madecassic acid, which are highly lipophilic compounds; and asiaticoside and madecassoside, which are hydrophilic compounds. These compounds were aimed at the target site to exert their activities. For these reasons, the development of a formulation that provides the high solubilizing effect of PRE and can carry the extract to the oral mucosa efficiently, such as a microemulsion system, is of interest.

Microemulsion systems are composed of oil, water, and surfactant, which feature as clear, isotropic, and thermodynamically stable mixtures. Microemulsions are attractive candidates as promising drug delivery systems because of their enhanced drug solubilization and stability, simplicity of preparation, and administration [6,21]. Medicinal products using microemulsion systems for delivery are marketed as research products such as capsules [22], patches [23,24], gels [22,25], drug delivery to the eye [25], lungs [26], and injection drugs [27]. The microemulsion can increase the solubility and absorption rate. In addition, it can increase the stability and bioavailability of the delivered drug, contributing to better efficacy of treatment [28]. Therefore, preparing natural extracts in microemulsion systems is suitable and interesting for developing the application as oral epithelial surface products.

In this study, the microemulsion systems were prepared using coconut oil or the combination of coconut oil and clove oil as an oil phase. Coconut oil is an edible oil that has been used in different emulsion systems, e.g., micro- and nanoemulsions [29,30,31]. It possesses antibacterial, anti-inflammatory, antioxidant, and wound-healing properties on the skin and oral mucosa [32,33]. Clove oil has been utilized in toothache and oral pain relief. It also exhibits antimicrobial, antioxidant, and anti-inflammatory effects [34,35]. These ingredients were aimed to aid in oral ulcer treatment. Tween 20 and Span 20 were chosen as surfactants because of their non-ionic features and safety for use in oral products [36]. Propylene glycol (PG) has been occasionally used as a cosurfactant to stabilize the microemulsion droplets [37].

This research aimed to develop an oral microemulsion spray containing PRE for healing mouth ulcers. The oral ulcer healing potential of the PRE was investigated and incorporated into the microemulsion system. The obtained microemulsion was then evaluated for its activities, including wound-healing activity on human dermal fibroblasts (HDF) and human gingival fibroblasts (HGF) and anti-inflammation activity by inhibition of nitric oxide (NO) production from macrophage cells to ensure its potential for oral ulcer treatment. The results of this study can be used in new herbal-based microemulsion product development for oral ulcer treatment.

## 2. Materials and Methods

### 2.1. Plant

Aerial parts of *C. asiatica* were collected from Songkhla Province, Thailand. A voucher specimen (specimen no. SKP 199 03 01 01) was authenticated by Associate Professor Dr. Panupong Puttarak and deposited at the herbarium of the Faculty of Pharmaceutical Sciences, Prince of Songkla University, Thailand. The plant was washed and dried at 60 °C for 24 h in a hot-air oven and reduced to powder using a grinder and a no. 45 sieve and kept in a closed container until use.

### 2.2. Chemicals and Reagents

The virgin coconut oil was purchased from Tropicana Oil Co., Ltd. (Nakhon Pathom, Thailand) and the clove oil was purchased from Chemipan Corporation Co., Ltd. (Bangkok, Thailand). Chemicals purchased from the PC drug center (Bangkok, Thailand) were propylene glycol, Tween 20, and Span 20. Standard asiatic acid, phosphate buffer saline (PBS), and 3-(4,5-dimethyl-2-thiazolyl)-2,5-diphenyl-2H-tetrazolium bromide (MTT) were obtained from Sigma-Aldrich (St. Louis, MO, USA). Standard madecassic acid, asiaticoside, and madecassoside were from Chengdu Biopurify Phytomedicals (Sichaun, China). Acetonitrile (HPLC grade) and ethanol (analytical grade) were from Labscan Asia (Bangkok, Thailand). Water was purified in a Milli-Q system (Millipore, Bedford, MA, USA). Dulbecco’s Modified Eagle Medium (DMEM) and fetal bovine serum were purchased from Invitrogen (Carlsbad, CA, USA).

### 2.3. Preparation of PRE

The PRE was prepared by the method previously described [17]. Briefly, the dried powder of *C. asiatica* was extracted by microwave-assisted extraction (MAE). The optimal conditions of MAE were extraction with absolute ethanol as solvent, an irradiation power of 600 W at 75 °C, four irradiation cycles (each cycle: 15 s power on and 30 s power off), and four extraction times. The pooled extracts were then dried in a vacuum. The crude ethanol extract was dissolved in 25% *v*/*v* ethanol in water. After filtering through the cotton wool, the solution was loaded into the macroporous resin (Diaion^®^ HP-20, Sigma-Aldrich, Germany) column and eluted with 25% *v*/*v* ethanol. The residue from the filtering step was dissolved in 50% *v*/*v* ethanol, and the solution was then loaded into the same column and eluted with 50% *v*/*v* ethanol and 75% *v*/*v* ethanol. The obtained pentacyclic triterpene-rich fractions were pooled and evaporated to dryness in a vacuum to obtain the PRE.

### 2.4. HPLC Quantitative Analysis of the Standardized Extract

HPLC quantitative analyses were performed for the *C. asiatica* crude extract, PRE, and microemulsion containing PRE following a previous method [17,19]. The different preparations of each type of sample are described below. A 15 mg sample of *C. asiatica* crude extract and PRE was accurately weighed, dissolved in methanol, and then adjusted to 10 mL in a volumetric flask. The microemulsion containing PRE (5 mL) was extracted with 15 mL of methanol, lightly shaken, soaked in a sonicate bath for 5 min. The obtained solution (1 mL) was then adjusted with 10 mL of methanol to achieve a sample solution with a concentration of 0.1 mg/mL. The prepared solution was filtered through a 0.45 µm PVDF membrane filter. The HPLC method was performed as previously described [17,19]. Briefly, the HPLC was carried out under gradient conditions using TSK gel ODS-100 V (250 × 4.6 mm). A mixture of acetonitrile and water in a gradient elution (ratio: 0–5 min, 20:80; 5–10 min, 30:70; 10–20 min, 65:35; 20–30 min, 70:30) was used as the mobile phase. The flow rate was 1 mL/min at the temperature of 30 °C with an injection volume of 20 µL. Pentacyclic triterpenes content in each sample was compared the area under the curve with the standard curve of pentacyclic triterpenes at 210 nm. Each sample was measured in triplicate. The calibration curves were established from the standards of madecassoside (MS), asiaticoside (AS), madecassic acid (MA), and asiatic acid (AA) at the concentration between 0.03 and 0.50 µg/mL (MS: y = 3 × 10^6^x − 8498.8, R² = 1; AS: y = 4 × 10^6^x + 3164.2, R² = 1; MA: y = 7 × 10^6^x + 8894.2, R² = 1; AA: y = 8 × 10^6^x − 1647.3, R² = 1).

### 2.5. Preparation of Pseudo-Ternary Phase Diagram of Microemulsion

In this research, the pseudo-ternary phase diagram was prepared to find the appropriate ratio of the components to prepare the microemulsion. The components of the microemulsion consisted of purified water as a water phase (W), coconut oil, or coconut oil:clove oil (1:1 or 2:1) as an oil phase (O), and the mixture of surfactants (SM); Tween 20, or Tween 20:Span 20 (1:1), or Tween 20:Span 20:Propylene glycol (PG) (2:1:1). These three components were mixed in different proportions according to the ratio on the phase diagram by vortex for 1 min. The portions that provided transparency were marked in the pseudo-ternary phase diagram to acknowledge the scope of the microemulsion formation.

### 2.6. Preparation of Microemulsion Containing PRE

After obtaining the microemulsion boundary from the pseudo-ternary phase diagram, the ratio of each component was selected for further preparation. The microemulsions containing PRE were prepared by mixing SM and PRE by stirring using a magnetic stirrer at ambient temperature to obtain 1%, 2.5%, and 5% *w*/*v* of PRE in the final preparations. Then O and W phases were added and mixed until a clear microemulsion was obtained.

### 2.7. Evaluation of the Physical Properties of the Microemulsion Base and Microemulsions Containing PRE

#### 2.7.1. Visual Observation

The obtained microemulsions were visually observed. The microemulsion should be transparent and flow when centrifuged by spinning 5000 rounds/min for 30 min, without the separation of any phases or particles.

#### 2.7.2. Type of Microemulsion Test

The microemulsion type was evaluated by dropping the water-soluble dye (Methylene blue or Ponceau 4R) into the formulations. If the dye dissolves, it is classified as an oil-in-water microemulsion, but it is classified as a water-in-oil microemulsion if the dye does not dissolve.

#### 2.7.3. Measurement of the pH

The microemulsions were placed in the test tube, and the pH of the microemulsions was tested by pH meter (Mettler LE409, Mettler Toledo, Columbus, OH, USA). The suitable pH range of the formulation should be between 6.2–7.4 to prevent irritation.

#### 2.7.4. Measurement of the Viscosity

The microemulsion viscosity was measured by Brookfield viscometer (DV-III ultra, Brookfield Engineering, Middleboro, MA, USA). The spindle number F (10K) was used. The spindle was rotated at 10 rotations per minute at 25 °C. The test was performed in triplicate.

#### 2.7.5. Measurement of the Conductivity

The conductivity of the microemulsion was measured by a conductivity meter (ST10C-A, OHAUS^®^, Parsippany, NJ, USA) at 25 °C. The test was performed in triplicate.

#### 2.7.6. Droplet Size

The mean droplet size of the microemulsion was determined by photon correlation spectroscopy using the Zetasizer (Malvern Instruments, Worcestershire, UK). Droplet size analysis was performed at 25 °C with an angle of detection at 90°. The droplet size of the formulations was obtained directly from the instrument [9].

### 2.8. Bioactivity Studies of Crude C. asiatica Extract, PRE, Pure Isolated Compounds, Microemulsion Base, and Microemulsion Containing PRE

The PRE, crude extract of *C. asiatica*, and pure isolated compounds (MS, AS, MA, and AA) were subjected to evaluation in cell proliferation assays and anti-inflammatory activities (anti-NO production assay). Cell proliferation, anti-inflammation, collagen production, and cell migration properties of the microemulsion base and microemulsion containing PRE were evaluated, as explained below.

#### 2.8.1. Cell Viability and Proliferation Assay

The PRE, *C. asiatica* crude extract, pure isolated compounds (MS, AS, MA, and AA), microemulsion base, and microemulsions containing PRE were tested with the cell proliferation method in human dermal fibroblast (HDF) cells [38]. HDF was purchased from ATCC^®^_,_ USA. In addition, the microemulsion base and microemulsions containing PRE were also tested in human gingival fibroblast (HGF) cells [38,39]. HGF was kindly provided by the Medical Science Research and Innovation Institute, Prince of Songkla University (from ATCC^®^, Manassas, WV, USA). The cell suspensions (1 × 10^4^ cells/well) were seeded in 96-well plates with Dulbecco’s Modified Eagle Medium (DMEM), which included 10% fetal calf serum. After 24 h, the new medium was replaced and then added with test substances. The test sample was dissolved in the DMSO to obtain concentrations of 1, 3, 10, 30, and 100 mg/mL, then achieving the desired concentration by mixing with the medium in each well. The cell without sample was used as a negative control. After incubating cells in a CO_2_ incubator at 37 °C for 24 h, the medium was removed and replaced by the fresh medium containing 10 µL of 3-(4,5-dimethylthiazolyl)-2,5-diphenyltetrazolium bromide (MTT) solution (5 mg/mL in phosphate buffer saline; PBS) and incubated at 37 °C for 4 h. Later, the medium was removed, and 200 µL of DMSO was added to dissolve the formazan from the living cell. The control consisted of the DMSO and media without samples. Cell proliferation was measured at a wavelength of 570 nm, and the percentage of cell proliferation was calculated compared with the control as follows.
Cell proliferation (%) = (Absorbance of sample)/(Absorbance of control) × 100(1)

#### 2.8.2. Migration Assay

The microemulsion base and microemulsions containing PRE were tested with the migration method on HDF and HGF cells [11,39]. The cell suspensions (1 × 10^5^ cells/well) were seeded in 24-well plates with DMEM medium that included 10% fetal calf serum, then incubated cells were placed in a CO_2_ incubator at 37 °C until cells spread all over the well. The pipette tip was used to scratch the monolayer cell and wash clear cells with PBS and add DMEM medium of 1 mL sample solution per well. The test sample was dissolved in the DMSO to achieve the desired concentration by mixing with the medium in each well. The control consisted of the DMSO and media without samples. The photo was taken with the microscope (Nikon ECLIPSE TS100, Japan) at 0 h; then cells were incubated in a CO_2_ incubator at 37 °C and repeated at 6, 12, 24, 48, and 72 h. The percentage of the decreased area was compared with 0 h using Image J.

#### 2.8.3. Anti-Inflammatory Activity Assay (Anti-NO Production)

The PRE, *C. asiatica* crude extract, pure isolated compounds (MS, AS, MA, and AA), microemulsion base, and microemulsion containing PRE were tested for anti-inflammatory activity, performed using the inhibition of NO production from murine macrophage cells (RAW 264.7). Raw 264.7 cells were purchased from the American Type Culture Collection (Manassas, VA, USA). This method was the same as Praparatana et al., 2022 [40]. They were cultured in Roswell Park Memorial Institute (RPMI) medium containing 10% fetal bovine serum (FBS), 0.1% sodium bicarbonate, 2 µg/mL glutamine, and penicillin-streptomycin solution (100 µg/mL) in a CO_2_ incubator at 37 °C. The cells were removed by trypsin-EDTA and suspended with a fresh RPMI medium. The cell suspension was allowed to adhere to the 96-well plate with 1 × 10^5^ cells/well for 60 min. After that, the RAW 264.7 cells were rinsed with PBS and replaced with 100 µL of RPMI medium containing lipopolysaccharide (LPS) to activate NO production. The sample and the positive control (standard indomethacin) were prepared using 1% DMSO in RPMI medium at various concentrations (3–100 µg/mL). One hundred milliliters of each test sample was added into the well and then incubated for 48 h. Griess reagent was used to determine the accumulation of NO production in the cell supernatant by spectrophotometry at 570 nm. IC_50_ values were determined using the plot of % inhibition and the concentration that could inhibit 50% of NO production (n = 4).
Inhibition (%) = [(A_control_ − A_sample_)/A_control_] × 100(2)

#### 2.8.4. Sircol Collagen Assay for Total Collagen in the Culture Medium

The microemulsion base and microemulsion containing PRE were tested with the Sircol collagen method on HDF and HGF cells to assess the ability to stimulate the collagen production of each formula. The total collagen assay was performed according to the report of Lareu and co-workers (2010) [41] with some modifications. In brief, the total collagen in the supernatant was assessed using the Sircol Soluble Collagen Assay kit (Biocolor, Northern Ireland). This total collagen assay is a quantitative dye-binding method to analyze collagens released into the culture medium in vitro. HDF/HGF cells were seeded in 96-well plates at 2 × 10^4^ cells/mL density. Cells were grown for 24 h and treated with or without samples for 24 h. At the end of incubation, 100 µL of solution from each well was transferred to a 1.5 mL centrifuge tube, and 500 µL Sircol dye reagent was added to each tube. The mixed contents were maintained and shaken at room temperature for 30 min. The tubes were then centrifuged at 10,000× *g* for 10 min, and the supernatants were discarded. The remaining pellets were cleaned with ethanol and then gently mixed with 500 µL alkali reagent until the precipitate was dissolved completely. The solutions were transferred to 96-well plates, and the plate was read using a spectrophotometer (Beckman Coulter, DTX 800, Austria) at 540 nm. The result was compared and expressed in standard collagen equivalents (mg/g).

### 2.9. Statistical Analysis

All data were subjected to statistical analysis using a sample *t*-test. *P*-values indicate statistical significance (* *p* < 0.05). Mean values ± SD were reported from three different experiments. Statistical analyses were carried out using the SPSS statistical software (SPSS, Inc., Chicago. IL, USA).

## 3. Results and Discussion

### 3.1. Preparation of Active Compound (Pentacyclic Triterpene-Rich Extract: PRE)

#### 3.1.1. Preparation of *C. asiatica* Crude Extracts and PRE

The 270 g dried powder of *C. asiatica* (Figure 1A) was extracted with the MAE method by absolute ethanol. The crude extracts (Figure 1B) (% yield of *C. asiatica* powder = 30.59% *w*/*w*) contained a total pentacyclic triterpene content of 8.23% *w*/*w*, containing glycoside 3.69% *w*/*w* and aglycone 4.53% *w*/*w* from the HPLC analysis. The PRE (% yield of *C. asiatica* crude extract = 4.06% *w*/*w*) (Figure 1C) showed the increase in the pentacyclic triterpene content up to 59.60% *w*/*w* (Table 1), containing glycoside 34.93% *w*/*w* and aglycone 24.67% *w*/*w*. PRE had the physical appearance of off-white powder. The preparation of the PRE was one of the standardized *C. asiatica* extract similar to the titrated extract of *Centella asiatica* (TECA) and ECa233, albeit with the different preparation method and pentacyclic contents [8,17,42,43]. PRE was prepared following a green extraction concept, which reduced the process and energy consumption, making it suitable for microemulsion enrichment in pentacyclic triterpenes for oral spray.

#### 3.1.2. Determination of the PRE, Crude *C. asiatica* Extract, and Pure Isolated Compound (MS, AS, MA, and AA) Bioactivities

In mouth ulcer patients, the inflammation can affect fibroblast proliferation. The proliferation effects of the crude extract and PRE on human dermal fibroblast (HDF) cells were evaluated to confirm the benefits of these extracts for mouth ulcer patients. The PRE, crude extract of *C. asiatica,* and pure isolated compounds were subjected to evaluation of their proliferation (cell proliferation assay) and anti-inflammatory activities (anti-NO production assay), as described below.

##### Cell viability and Proliferation Assay

The cell viability and proliferation of all tested compounds at 1, 3, 10, 30, and 100 μg/mL are presented in Table 2. It was found that the enhancement effect on HDF cell proliferation was achieved by the MTT method. The results showed that PRE exhibited a potent proliferative effect (165.67 ± 1.95%) at 10 μg/mL and decreased when using higher concentrations at 30 and 100 μg/mL (148.51 ± 1.96% and 117.80 ± 1.16%, respectively). These results may be due to the cytotoxicity, which was also observed in the previous study where high doses of pentacyclic triterpenes could decrease cell viability [42]. The PRE showed a higher proliferation property than the crude extract and pure isolated compounds (MS, AS, MA, and AA) owing to the increase in pentacyclic triterpene content in the PRE. Higher content of pentacyclic triterpenes, better biological activities, dose adjustment, and physical appearance were observed in PRE when compared to the crude extract. Our findings, in line with previous reports on the chemical complexity of the extract, were that it showed a greater advantage over pure isolated compounds in terms of the wide range of pharmacological activities and lower toxicity, and it was also cost-effective [19].

##### Anti-Inflammatory Activity of PRE, Crude Extract from C. asiatica, Pure Isolated Compounds (MS, AS, MA, and AA), and Indomethacin

Evaluation of the anti-inflammatory activity of PRE, crude extract from *C. asiatica*, and pure isolated compounds via inhibitory effect against NO production revealed that PRE possessed a satisfactory NO inhibitory effect with an IC_50_ value of 20.59 ± 3.48 μg/mL (Table 3). The anti-inflammatory activity of PRE was higher than that of *C. asiatica* crude extract, MA, MS, AS, and a standard drug (indomethacin) but less than that of AA (IC_50_ 5.07 ± 0.23 μg/mL). Furthermore, non-cytotoxicity (macrophage cell) was found in PRE with % cell viability >80% at 1, 3, 10, 30 and 100 µg/mL.

Considering the cell proliferation and anti-inflammatory activity results, the PRE had a better wound-healing effect than the crude extract and three pure isolated compounds (MS, AS, and MA). Our findings were aligned with previous reports on the anti-inflammatory activity, in which AA and MA showed a more potent inhibitory activity against lipopolysaccharide-induced NO and PGE2 production than AS and MS [26]. Moreover, there have been reports that PRE has antibacterial activity against *Streptococcus* spp., and it is also suitable for anti-inflammation and antioxidant applications in treating oral mucositis [17,43]. From the results, PRE was considered a suitable active ingredient in the formulation of an oral spray microemulsion as PRE exerted good proliferation and anti-inflammatory activity.

### 3.2. Evaluation of the Microemulsion Base and Microemulsions Containing PRE in Various Concentrations

In this study, six pseudo-ternary phase diagrams (M1-M6) of different compositions of O and SM phases were produced to determine the appropriate ratio of W, O, and SM to provide a transparent range of the microemulsion.

The phase diagram M1-M6, as represented in Figure 2, was constructed for the variable weight ratio of SM (Tween20, Span20, and PG) and O (the mixture of coconut oil and clove oil). The microemulsion region is presented in the diagram, and the rest of the region represents the formation of turbid emulsion or phase separation. Therefore, the formulations of M1, M2, M3, M4, M5, and M6 show the 3, 1, 2, 4, 5, and 6 points of transparent microemulsions (Appendix A), respectively, on the phase diagrams (Figure 2).

The M6 with the weight ratio of Tween 20: Span 20 (2:1) and coconut oil: clove oil (1:1) showed a wide concentration range forming the microemulsion (yellow star). Thus, M6 was selected for further development. Every transparent microemulsion spot on the M6 diagram was evaluated for its physical properties (Table 4).

The physical properties of the microemulsion bases (F1–F6) prepared from the different ratios of W, O, and SM presented in the pseudo-ternary phase diagram plotted in M6 are demonstrated in Table 4. The microemulsion area of M6 was observed with a relatively high percentage of SM (50–80%) compared to the O (10–30%) and W (10–30%) content. The droplet size of F1–F6 was less than 100 nm, which could be classified as a microemulsion, and its pH range of 5–7 can be used orally. All formulations had low viscosity and were defined as w/o microemulsion according to the drop dilution test. In this study, the suitable microemulsion base was selected based on the lowest SM content possible to avoid irritation on the oral mucosa, suitable for the spraying of the aerosol and the smallest droplet size. The microemulsion base selected for the formulation development was composed of W:O:SM 2:2:6 (F4). Although the F5 ratio of W:O:SM 3:2:5 contained the lowest ratio of SM, lowest viscosity, and smallest size, it is at the border of the diagram and can be disturbed by a slight variation in the composition. The plot of F4 on the pseudo-ternary phase diagram M6 is shown in Figure 3 (yellow star). Considering that the physicochemical properties of the active constituents in the PRE, AA, and MA formulations were highly lipophilic compounds, while AS and MS were hydrophilic compounds, the selected microemulsion must be able to dissolve all the active constituents in the system. The F4 formula contained equal amounts of the water and oil phases and a suitable ratio of SM to solubilize the PRE, dissolving all the extracts and achieving a transparent microemulsion suitable for a spray.

#### 3.2.1. Formulation of Microemulsions Containing PRE

Because of the limitations of previous research, the solution recipe containing PRE achieved only 0.1% *w*/*v* owing to the low water solubility of some compounds in the extracts, low absorption rate, and unfavorable scent and taste [17,44]. This research aimed to develop an oral spray microemulsion that addressed these problems. In treating minor recurrent aphthous ulcers, a 0.05% *w*/*v* standardized extract of *C. asiatica* has previously demonstrated a wound-healing effect [8]. Concentrations of 1%, 2.5%, and 5% were then selected to explore if a higher PRE concentration in the formula could possibly be maintained in the microemulsion system. In addition, the effect of PRE concentrations on their wound-healing (HDF and HGF) and anti-inflammation activities was determined to select the suitable concentration for oral spray. Varying concentrations of 0.00 (base), 1.0%, 2.5%, and 5.0% PRE in a microemulsion base were prepared to confirm the hypothesis that all formulas are effective in treating oral ulcers.

The formulations containing 1%, 2.5%, and 5% PRE presented nearly the same properties in physical appearance, droplet size, pH, viscosity, and conductivity compared with a microemulsion base (Table 5). The microemulsion containing PRE was homogeneous and transparent. After mixing PRE (off-white color powder) in the microemulsion base, the light yellowish to dark brown color was obtained in which the intensity of the color corresponded to the amount of PRE (Appendix A). The droplet size was in the range of 26–46 nm, viscosity and conductivity were slightly increased, and the pH did not show a tendency to change. The results indicated that all microemulsion formulations were suitable for an oral spray for oral ulcer treatment. The results are shown in Table 5.

#### 3.2.2. Evaluation of the Biological Activity of the Microemulsion Base and Microemulsions Containing PRE at Various Concentrations

Anti-inflammatory activity and wound-healing properties of the microemulsion base and microemulsions containing PRE were studied. Various properties including proliferation (cell viability or proliferation assay), migration activities (in vitro wound-healing scratch assay), total collagen in the culture medium (Sircol collagen assay), and anti-inflammatory activity assay were evaluated. The best formulation demonstrating good wound-healing and potent anti-inflammatory activity was selected as an oral ulcer treatment spray.

##### Wound-Healing Property

The wound-healing property was evaluated by the proliferation and migration activities in HDF and HGF cells on skin and oral wounds. The HDF and HGF cells were treated with different concentrations at 1, 3, 10, 30, and 100 µg/mL of the microemulsion base and microemulsions containing 1%, 2.5%, and 5% PRE.

##### Proliferation of HDF Cells

It was found that the enhancement effect on HDF cell proliferation was achieved by the MTT method. The results showed that microemulsions containing 1% and 2.5% PRE exhibited a potent proliferative effect (123.91–136.30%) at 3 and 10 μg/mL, while the microemulsion containing 5% PRE acted as a proliferative inducer (108.54 ± 4.67%) at low concentrations (1 μg/mL), and the microemulsion base showed the lowest percent proliferation (<100%) on HDF cells. The microemulsions containing 1%, 2.5%, and 5% PRE showed the highest proliferation property at concentrations of 10, 3, and 1 μg/mL, respectively. All samples had a low proliferative effect at higher concentrations. Our results suggested that the pentacyclic triterpene content in each formulation played an important role in cell proliferation. This was consistent with previous test results on the cell proliferation of PRE that showed the best activity at the optimum concentration [42,43]. The cell viability of samples at 0.3, 1, 3, 10, and 30 μg/mL is presented in Table 6. It should be noted that all microemulsions containing PRE stimulated the growth of HDF cells.

##### Migration Assay (Scratch assay) of HDF Cells

Cell migration was evaluated by scratch assay on HDF cells with the microemulsion base and microemulsions containing 1%, 2.5%, and 5% PRE. The results indicated that the percentage of cell movements of the microemulsions containing 1% and 2.5% PRE had the highest migration at 24 h (100% and 94.5%, respectively) as shown in Table 7 (Appendix A). The microemulsions containing 1% and 2.5% PRE displayed higher migration than that with 5% PRE (68.69%) and the microemulsion base (70.92%) at 24 h. Interestingly, most tested samples showed over 100% migration activity at 48 h, which was higher than that of the control group (81.88%). The results indicated that the microemulsion containing PRE had the remedial effect to close the wound distance.

This study evaluated the wound-healing activity in HDF cells of the microemulsion base and microemulsions containing PRE at various concentrations by proliferation and migration tests. It was found that the formulation with good wound-healing activity was the microemulsion containing 1% PRE, providing the highest proliferation (136.30 ± 3.93% at a concentration of 10 μg/mL) and migration activities (100.00 ± 0.00 at 24 h) in HDF cells when compared with other formulations.

##### Cell Proliferation of HGF Cells

HGF cells were evaluated as they are cryopreserved human oral fibroblasts derived from adult gingival tissue. They can be used to verify the healing ability of the formulation on oral ulcers. In addition, many products that claimed to be oral wound-healing may not be studied in HGF cells. Therefore, this research was tested in both HDF and HGF cells to confirm the effectiveness of the microemulsion.

The proliferation rates of HGF cells after being treated with microemulsions containing 1%, 2.5%, and 5% PRE were different. The microemulsion containing 1% PRE showed the highest proliferation property at the concentration of 10 μg/mL (152.65 ± 3.48%), the microemulsion containing 2.5% PRE showed the highest proliferation property at a concentration of 3 μg/mL (144.89 ± 1.96%), and the microemulsion containing 5% PRE showed the highest proliferation property at a concentration of 1 μg/mL (141.22 ± 8.03%). The proliferative effect was decreased with higher concentrations of microemulsions. The results were consistent with the previous test results on the HDF cell proliferation of PRE. The microemulsion base exhibited the lowest percent proliferation when compared with the microemulsions containing PRE. All test results are shown in Table 8.

##### Migration Assay (Scratch Assay) of HGF cells

The percent cell movement of the microemulsion containing 1% PRE showed the highest HGF cell migration at 24 h (100.00%), followed by the microemulsion containing 2.5% PRE (92.31 ± 4.66%). The microemulsion containing 5% PRE resulted in only 61.85 ± 6.65% cell migration. In comparison, the control group at 24 h achieved only 55.10%. Moreover, all tested concentrations exhibited higher migration activity on HGF cells when compared to the control. Cell migration rates of the microemulsions containing 1%, 2.5%, and 5% PRE and the microemulsion base are shown in Table 9 (Appendix A).

The formulation with good wound-healing activity was the microemulsion containing 1% PRE, with the highest proliferation (152.65 ± 3.48% at a concentration of 10 μg/mL) and migration activities (100.00 ± 0.0% at 24 h) in HGF cells when compared with the others. The result was consistent with previous test results on HDF cells. All experiments above involved both types of fibroblast cells and supported the use of a microemulsion containing 1% PRE to be used in the oral wound-healing spray.

##### Sircol Collagen Assay for Total Collagen in the Culture Medium

Numerous studies have documented the action of triterpenes from *C. asiatica,* such as MS, MA, and AA, on the synthesis of collagen related to the wound-healing process [45,46,47]. The Sircol collagen assay experiments revealed that the substances can be added to the microemulsion formula to stimulate collagen production. The collagen content in the medium of fibroblasts incubated with the test sample was measured. Collagen production on HDF and HGF cells by the microemulsion base and microemulsions containing PRE at various concentrations were evaluated and compared with the control. For HDF cells, it was found that the microemulsion containing 1% PRE exhibited the highest collagen content of 55.80 ± 0.54 μg/mL, followed by the base (53.80 ± 1.15 μg/mL), 2.5% PRE (53.10 ± 1.91 μg/mL), and 5% PRE (51.50 ± 0.94). For HGF cells, it was found that the collagen content of the microemulsion containing 1% PRE also showed the highest collagen content at 52.80 ± 0.54 μg/mL, followed by 5% PRE (50.10 ± 0.94 μg/mL), 2.5% PRE (49.60 ± 1.91 μg/mL), and the base (46.80 ± 0.54 μg/mL). Combined with the statistical analysis, it was concluded that all formulations tested revealed slightly significant differences better than the control (p<0.05). However, all formulations tested did not show significant differences between each formulation. Collagen production in both cells was not different between the base and various concentrations. This may be due to the coconut oil in the preparation, which also promoted collagen production [48,49]. The results revealed that the combination of PRE and coconut oil facilitated collagen production in fibroblast cells. The results obtained are shown in Table 10.

##### Anti-Inflammatory Activity

The anti-inflammatory activity of the microemulsion base and microemulsions containing 1%, 2.5%, and 5% PRE, via inhibitory effect against NO production, were also tested. The results revealed that the microemulsion containing PRE could increase the NO inhibitory effect in a dose-dependent manner. The microemulsion containing 5% PRE possessed the highest NO inhibitory effect with an IC_50_ value of 10.94 μg/mL, followed by those containing 2.5% and 1% PRE (IC_50_ at 29.22 and 33.82 μg/mL, respectively). The anti-inflammatory activity of the microemulsion containing 1% PRE was slightly lower than standard indomethacin. Moreover, the microemulsion containing 5% PRE exhibited higher anti-inflammatory activity than the standard indomethacin (IC_50_ at 25.06 ± 2.63 μg/mL). At the same time, the microemulsion base did not affect NO production with an IC_50_ value of 83.44 μg/mL (Table 11). The inhibition effects of the microemulsions containing PRE shown were correlated to the concentrations of PRE accordingly.

Considering the results of cell proliferation and cell migration of the microemulsion base and microemulsions containing 1.0%, 2.5%, and 5.0% PRE, the microemulsions containing PRE had a better wound-healing effect than the microemulsion base according to the optimum concentration of PRE. The microemulsion containing 1% PRE had the highest proliferation (136.30 - 152.65% at a concentration of 10 μg/mL) and migration activities (100.00 ± 0.0 at 24 h) in HDF and HGF cells when compared with other formulations. Additionally, the microemulsion containing 1% PRE increased collagen production in HDF and HGF cells when compared to the control (blank). This formulation showed the best oral wound-healing effect. It should be noted that while the anti-inflammatory activity by inhibiting NO production of the microemulsion containing 5% PRE was higher than the 1% PRE, it still exhibited good anti-inflammatory activity. Thus, the microemulsion with 1% PRE was chosen as the most suitable oral spray formulation.

## 4. Conclusions

The preparation of a standardized *C. asiatica* extract (PRE) for oral ulcer treatment was performed in this study. The PRE was then evaluated to confirm the benefits of these extracts for mouth ulcers. Considering the cell proliferation and anti-inflammatory activity results, the off-white powder PRE had a better wound-healing effect than the crude extract and three pure isolated compounds (MS, AS, and MA). Conclusively, PRE is suitable for the preparation of an oral microemulsion spray for oral ulcer treatment.

Through the pseudo-ternary phase diagram approach, we found the most suitable formula to be F4, which was classified as microemulsions with the nanosized droplet, optimal oral pH, low viscosity, and suitable for spraying. F4 was selected as the microemulsion base of the oral spray containing the PRE preparation.

The microemulsions containing 1%, 2.5%, and up to 5% PRE were prepared. The most suitable oral spray formulation containing 1% PRE had the highest proliferation and migration activities, induced collagen production, and exhibited good anti-inflammatory activity.

This study achieved the objectives of formulating an oral microemulsion spray for oral ulcer treatment. Further clinical efficacy and safety studies should be conducted before product registration.

## Figures and Tables

**Figure 1 pharmaceutics-14-02531-f001:**
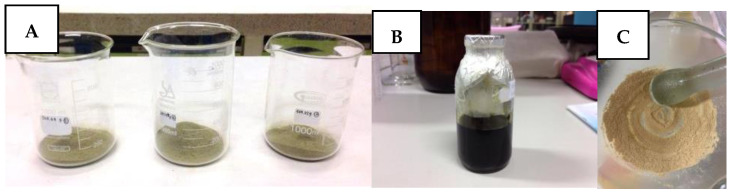
The dried powder (**A**), the crude extract (**B**), and PRE (**C**) from *C. asiatica*.

**Figure 2 pharmaceutics-14-02531-f002:**
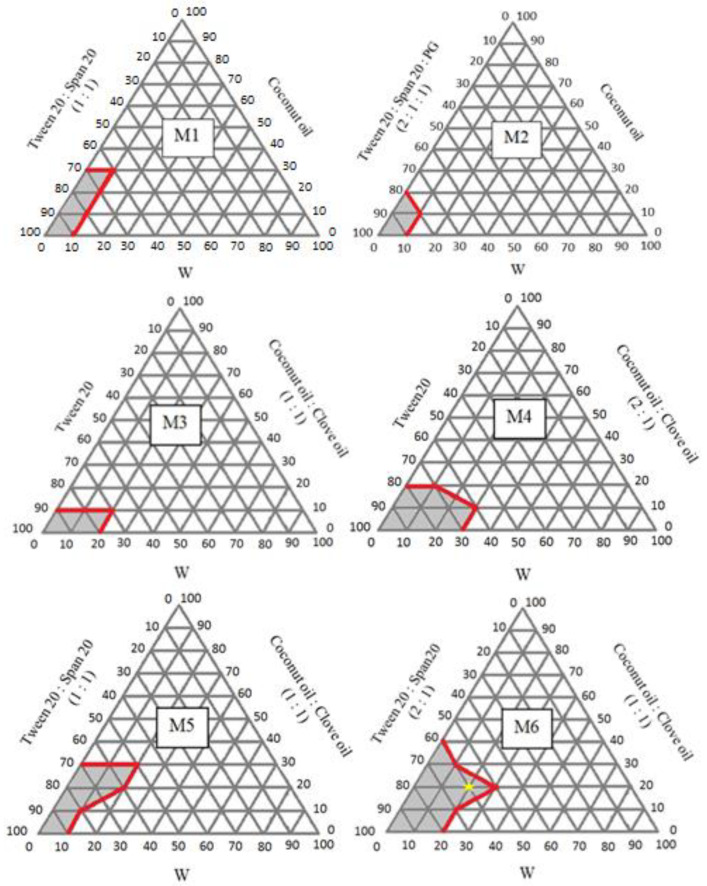
Pseudo-ternary phase diagrams M1–M6.

**Figure 3 pharmaceutics-14-02531-f003:**
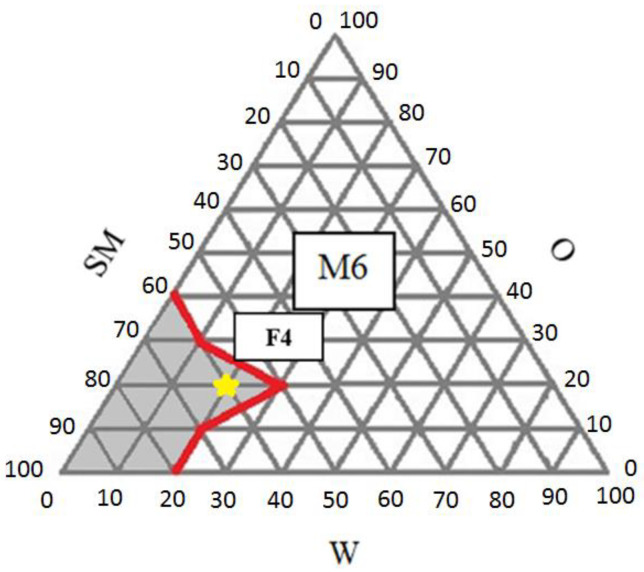
Plot of F4 on pseudo-ternary phase diagram M6 (W = water, O = oil, and SM = surfactant mixture).

**Table 1 pharmaceutics-14-02531-t001:** Pentacyclic triterpene content of PRE and crude extracts from *C. asiatica*.

Samples	Yield (mg/g)	Pentacyclic Triterpene Content (% *w*/*w*) (Mean ± SD)						
		MS	AS	MA	AA	Glycoside	Aglycone	Total
Crude extract	30.59	2.22 ± 0.04	1.47 ± 0.20	2.90 ± 0.44	1.63 ± 0.04	3.70 ± 0.20	4.53 ± 0.41	8.23 ± 0.22
PRE	1.24	13.43 ± 0.22	21.50 ± 0.48	17.94 ± 0.20	6.73 ± 0.49	34.93 ± 0.18	24.67 ± 0.69	59.60 ± 0.76

MS: Madecassoside; AS: Asiaticoside; MA: Madecassic acid; AA: Asiatic acid.

**Table 2 pharmaceutics-14-02531-t002:** Effect of PRE, crude extract, and pure isolated compounds (MS, AS, MA, and AA) on HDF cell proliferation.

Sample	% Proliferation				
	1 µg/mL	3 µg/mL	10 µg/mL	30 µg/mL	100 µg/mL
PRE	147.97 ± 1.66 *^,#^	153.93 ± 1.11 *^,#^	165.67 ± 1.95 *^,#^	148.51 ± 1.96 *^,#^	117.8 ± 1.16 *^,#^
Crude extract	87.10 ± 4.10 *	90.54 ± 3.05 *	97.73 ± 3.26	98.75 ± 1.15	106.72 ± 1.80 *
MS	103.60 ± 0.73 *	113.00 ± 0.73 *	98.46 ± 0.73	99.42 ± 1.73	91.06 ± 1.18 *
AS	104.57 ± 0.46 *	108.94 ± 0.09 *	117.70 ± 0.27 *	99.42 ± 0.09	86.87 ± 0.36 *
MA	99.29 ± 0.23	105.77 ± 0.62 *	94.29 ± 0.85 *	90.26 ± 0.69 *	36.72 ± 1.00 *
AA	101.20 ± 0.15	103.43 ± 0.23 *	102.45 ± 1.00	35.26 ± 0.31 *	31.61 ± 0.23 *
Control	100.00 ± 1.98	100.00 ± 1.98	100.00 ± 1.98	100.00 ± 1.98	100.00 ± 1.98

Results are represented as the mean ± SD of three independent determinations (N = 3); values in the PRE, crude extract, the pure isolated compounds of *C. asiatica,* and control column followed by symbol (*) are significantly different (*p* < 0.05); and values in the crude extract and PRE column followed by symbol (#) are significantly different (*p* < 0.05).

**Table 3 pharmaceutics-14-02531-t003:** Nitric oxide inhibitory activities of PRE, crude extract from *C. asiatica*, pure isolated compounds (MS, AS, MA, and AA), and indomethacin.

Sample	% Cell Viability					IC_50_ (μg/mL)
	1 µg/mL	3 µg/mL	10 µg/mL	30 µg/mL	100 µg/mL	
PRE	95.86 ± 2.85	105.28 ± 1.85	103.71 ± 2.24	104.48 ± 1.62	127.37 ± 1.22	20.59 ± 3.48 ^b^
Crude extract	88.14 ± 0.47	89.55 ± 3.87	96.62 ± 2.25	98.80 ± 3.32	106.59 ± 3.42	64.61 ± 0.04 ^g^
MS	77.53 ± 0.14	75.44 ± 0.33	67.15 ± 0.19	58.53 ± 0.14	56.21 ± 0.12	56.21 ± 0.12 ^f^
AS	102.44 ± 0.1	99.32 ± 0.20	85.59 ± 0.07	66.71 ± 0.14	52.23 ± 0.17	52.23 ± 0.17 ^e^
MA	106.53 ± 0.32	103.50 ± 0.07	93.07 ± 0.23	52.77 ± 0.17	50.99 ± 0.11	50.99 ± 0.11 ^d^
AA	109.80 ± 0.10	105.78 ± 0.12	100.71 ± 0.15	8.35 ± 0.20	5.07 ± 0.23	5.07 ± 0.23 ^a^
Indomethacin	82.75 ± 0.57	75.27 ± 3.28	82.10 ± 1.42	78.51 ± 4.20	76.52 ± 3.45	32.45 ± 2.63 ^c^

Results are represented as the mean ± SD of three independent determinations (N = 3); values in the column followed by a different letter superscript (a–g) are significantly different (*p* < 0.05); and values having the same superscript are not statistically significant.

**Table 4 pharmaceutics-14-02531-t004:** Physical properties of the transparent microemulsion in the M6 diagram.

Formulations	SM (%)	O (%)	W (%)	Viscosity (cP)	Droplet Size (nm)	pH	Type
F1	80	10	10	212 ± 1	24 ± 4	5.74 ± 0.03	w/o
F2	70	10	20	176 ± 3	18 ± 2	6.37 ± 0.02	w/o
F3	70	20	10	109 ± 1	94 ± 2	6.05 ± 0.01	w/o
F4	60	20	20	97 ± 1	38 ± 3	6.11 ± 0.01	w/o
F5	50	20	30	76 ± 2	8 ± 1	6.16 ± 0.03	w/o
F6	60	30	10	124 ± 2	61 ± 3	6.54 ± 0.04	w/o

**Table 5 pharmaceutics-14-02531-t005:** Physical properties of the microemulsion base and microemulsions containing PRE at various concentrations.

Parameters	Base	1% PRE	2.5% PRE	5% PRE
Appearance	Clear liquid	Clear liquid	Clear liquid	Clear liquid
Color	Light yellow	Yellow	Orange	Brown
Droplet size (nm)	38 ± 3	26 ± 2	27 ± 2	46 ± 3
pH values	6.11 ± 0.01	6.02 ± 0.02	6.24 ± 0.04	6.79 ± 0.02
Viscosity (cP)	97 ± 1	97 ± 2	98 ± 2	101 ± 3
Conductivity (μS)	12.30 ± 0.05	13.10 ± 0.05	13.80 ± 0.10	13.90 ± 0.05
Separation	No	No	No	No

Base = microemulsion base, 1%, 2.5%, and 5% PRE = microemulsions containing 1%, 2.5%, and 5% PRE, respectively.

**Table 6 pharmaceutics-14-02531-t006:** Effect of the microemulsion base and microemulsions containing PRE at various concentrations on HDF cell proliferation.

Samples	% Proliferation				
	1 µg/mL	3 µg/mL	10 µg/mL	30 µg/mL	100 µg/mL
base	86.68 ± 4.84 *	94.57 ± 5.14	90.38 ± 7.05 *	89.74 ± 3.18 *	86.51 ± 3.60 *
1% PRE	97.79 ± 3.89 ^#^	122.28 ± 2.33 *^,#^	136.30 ± 3.93 *^,#^	102.63 ± 6.44 ^#^	93.28 ± 3.48 *^,#^
2.5% PRE	96.27 ± 8.80 ^#^	123.91 ± 3.96 *^,#^	114.90 ± 4.28 *^,#^	87.26 ± 3.75 *	84.78 ± 5.85 *
5% PRE	108.54 ± 4.67 *^,#^	81.83 ± 7.41 *^,#^	81.52 ± 6.96 *	78.41 ± 3.10 *^,#^	72.67 ± 7.99 *^,#^
Control	100.00 ± 1.31	100.00 ± 1.31	100.00 ± 1.31	100.00 ± 1.31	100.00 ± 1.31

Results are represented as the mean ± SD of three independent determinations (N = 3); values in the base, 1% PRE, 2.5% PRE, 5% PRE, and control column followed by symbol (*) are significantly different (*p* < 0.05); and values in the base, 1% PRE, 2.5% PRE, and 5% PRE column followed by symbol (#) are significantly different (*p* < 0.05) (base = microemulsion base, 1% PRE, 2.5% PRE, and 5% PRE = microemulsions containing 1%, 2.5%, and 5% PRE, respectively).

**Table 7 pharmaceutics-14-02531-t007:** Effect of the microemulsion base and microemulsions containing PRE at various concentrations on HDF cell migration.

Samples	% Migration Rate of Cells						
	0 h	6 h	12 h	18 h	24 h	36 h	48 h
Base	0.00 ± 0.00	20.52 ± 5.84	35.61 ± 4.66	55.47 ± 3.49 *	70.92 ± 4.65 *	86.00 ± 2.65 *	100.00 ± 0.00 *
1% PRE	0.00 ± 0.00	30.28 ± 3.65 *^,#^	62.82 ± 3.22 *^,#^	80.15 ± 2.45 *^,#^	100.00 ± 0.00 *^,#^	100.00 ± 0.00 *^,#^	100.00 ± 0.00 *
2.5% PRE	0.00 ± 0.00	27.48 ± 2.65 *	41.17 ± 1.63 *	76.86 ± 1.24 *^,#^	94.54 ± 4.66 *^,#^	100.00 ± 0.00 *^,#^	100.00 ± 0.00 *
5% PRE	0.00 ± 0.00	16.12 ± 4.65 ^#^	30.31 ± 1.06 *	50.95 ± 7.31 *	68.69 ± 6.65 *	80.58 ± 5.65 *	100.00 ± 0.00 *
Control	0.00 ± 0.00	13.96 ± 3.55	22.83 ± 3.46	38.23 ± 4.27	55.15 ± 3.29	71.00 ± 2.65	81.88 ± 1.56

Results are represented as the mean ± SD of three independent determinations (N = 3); values in the base, 1% PRE, 2.5% PRE, 5% PRE, and control column followed by symbol (*) are significantly different (p<0.05); and values in the base, 1% PRE, 2.5% PRE, and 5% PRE column followed by symbol (#) are significantly different (*p* < 0.05) (base = microemulsion base, 1% PRE, 2.5% PRE, and 5% PRE = microemulsions containing 1%, 2.5%, and 5% PRE, respectively).

**Table 8 pharmaceutics-14-02531-t008:** Effect of the microemulsion base and microemulsions containing PRE at various concentrations on HGF cell proliferation.

Samples	% Proliferation				
	1 μg/mL	3 μg/mL	10 μg/mL	30 μg/mL	100 μg/mL
Base	111.42 ± 2.24 *	114.96 ± 3.61 *	114.42 ± 2.67 *	115.23 ± 0.52 *	110.47 ± 1.63 *
1%	133.87 ± 2.47 *^,#^	139.86 ± 4.49 *^,#^	152.65 ± 3.48 *^,#^	143.53 ± 2.52 *^,#^	120.81 ± 2.70 *^,#^
2.5%	137.08 ± 6.02 *^,#^	144.89 ± 1.96 *^,#^	141.49 ± 3.11 *^,#^	130.61 ± 2.88 *^,#^	110.34 ± 3.70 *
5%	141.22 ± 8.03 *^,#^	133.33 ± 4.42 *^,#^	134.01 ± 1.74 *^,#^	119.59 ± 4.58 *	100.13 ± 2.69 ^#^
Control	100.00 ± 1.43	100.00 ± 1.43	100.00 ± 1.43	100.00 ± 1.43	100.00 ± 1.43

Results are represented as the mean ±SD of three independent determinations (N = 3); values in the base, 1% PRE, 2.5% PRE, 5% PRE, and control column followed by symbol (*) are significantly different (*p* < 0.05); and values in the base, 1% PRE, 2.5% PRE, and 5% PRE column followed by symbol (#) are significantly different (*p* < 0.05) (base = microemulsion base, 1% PRE, 2.5% PRE, and 5% PRE = microemulsions containing 1%, 2.5%, and 5% PRE, respectively).

**Table 9 pharmaceutics-14-02531-t009:** Effect of the microemulsion base and microemulsions containing PRE at various concentrations on HGF cell migration.

Samples	% Migration Rate of Cells						
	0 h	6 h	12 h	18 h	24 h	36 hs	48 h
Base	0.00 ± 0.00	27.52 ± 5.84 *	42.61 ± 4.66 *	56.47 ± 3.49 *	76.92 ± 4.65 *	100.00 ± 0.00 *	100.00 ± 0.00
1% PRE	0.00 ± 0.00	30.28 ± 3.65 *	62.82 ± 3.22 *^,#^	78.15 ± 2.45*^,#^	100.00 ± 0.00 *^,#^	100.00 ± 0.00 *	100.00 ± 0.00
2.5% PRE	0.00 ± 0.00	21.48 ± 2.65 *	41.17 ± 1.63 *	68.86 ± 1.24 *^,#^	92.31 ± 4.66*^,#^	100.00 ± 0.00 *	100.00 ± 0.00
5% PRE	0.00 ± 0.00	16.12 ± 4.65 *^,#^	40.31 ± 1.06 *	50.95 ± 7.31 *	61.85 ± 6.65 ^#^	100.00 ± 0.00 *	100.00 ± 0.00
Control	0.00 ± 0.00	11.96 ± 3.55	21.83 ± 3.46	36.23 ± 4.27	52.15 ± 3.29	81.88 ± 2.65	100.00 ± 0.00

Results are represented as the mean ±SD of three independent determinations (N = 3); values in the base, 1% PRE, 2.5% PRE, 5% PRE, and control column followed by symbol (*) are significantly different (*p* < 0.05); and values in the base, 1% PRE, 2.5% PRE, and 5% PRE column followed by symbol (#) are significantly different (*p* < 0.05) (base = microemulsion base, 1% PRE, 2.5% PRE, and 5% PRE = microemulsions containing 1%, 2.5%, and 5% PRE, respectively).

**Table 10 pharmaceutics-14-02531-t010:** Collagen content in the culture medium determined by the Sircol Collagen Assay after 24 h of incubation of HDF and HGF fibroblasts with the test sample at a concentration of 10 µg/mL.

Type of Sample	Collagen Content (μg/mL)	
	HDF	HGF
Base	53.80 ± 1.15	46.80 ± 0.54 *
1% PRE	55.80 ± 0.54 *	52.80 ± 0.54 *
2.5% PRE	53.10 ± 1.91	49.60 ± 1.91
5% PRE	51.50 ± 0.94 *	50.10 ± 0.94
Control cells	54.00 ± 0.84	49.00 ± 0.84

Results are represented as the mean ± SD of three independent determinations (N = 3); values in the base, 1% PRE, 2.5% PRE, 5% PRE, and control cells followed by symbol (*) are significantly different (*p* < 0.05) (base = microemulsion base; 1% PRE, 2.5% PRE, and 5% PRE = microemulsions containing 1%, 2.5%, and 5% PRE, respectively).

**Table 11 pharmaceutics-14-02531-t011:** Nitric oxide inhibitory activities of the microemulsion base and microemulsions containing PRE.

Sample	% Cell Viability					NO Inhibition IC_50_ μg/mL
	1 µg/mL	3 µg/mL	10 µg/mL	30 µg/mL	100 µg/mL	
Base	84.08 ± 4.48	77.81 ± 4.29	95.84 ± 4.96	79.29 ± 3.40	68.13 ± 1.65	83.44 ± 12.10 ^d^
1% PRE	93.00 ± 0.68	107.18 ± 0.14	119.40 ± 2.07	109.67 ± 3.78	104.59 ± 3.72	33.82 ± 4.88 ^c^
2.5% PRE	110.72 ± 4.73	109.89 ± 4.03	118.69 ± 4.23	111.34 ± 3.26	99.37 ± 2.53	29.22 ± 1.59 ^c^
5% PRE	111.10 ± 4.07	120.24 ± 4.88	127.84 ± 2.54	113.30 ± 2.93	112.65 ± 3.26	10.94 ± 1.63 ^a^
Indomethacin	82.75 ± 0.57	75.27 ± 3.28	82.10 ± 1.42	78.51 ± 4.20	76.52 ± 3.45	25.06 ± 2.63 ^b^

Results were represented as the mean ± SD of three independent determinations, values in the column followed by a different letter superscript (a–d) are significantly different (*p* < 0.05), and values having the same superscript are not statistically significant.

## Data Availability

Not applicable.

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
