# Peer review of "Development of Oral Microemulsion Spray Containing Pentacyclic Triterpenes-Rich *Centella asiatica* (L.) Urb. Extract for Healing Mouth Ulcers"

_pharmaceutics, 2022, doi:10.3390/pharmaceutics14112531_

Round 1

Reviewer 1 Report

This manuscript describes the evaluation of anti-inflammatory activity and HDF proliferation assays of C. asiatica crude and PRE extracts to assess their effectiveness for wound healing. Then a microemulsion was prepared and adequately characterized for the incorporation of the PRE. Different assays were performed to conclude that the microemulsion containing 1% PRE is promising for oral ulcer treatment.

The article presents the study conducted in detail but needs some corrections to be considered for publication. Regarding the experimental design, the only flaw, from my point of view, was that they did not carry out a PRE release study to try to explain the activity results obtained.

The authors can find below my considerations and suggestions in detail.

ABSTRACT – must be deeply revised

Line 18: the described method increased the pentacyclic triterpene content to 59.6% - that increase is concerning what? Revise it
Suggestion: “This method furnished a pentacyclic content of 59.6% w/w…”

Line 19: What do you mean by “satisfactory physical appearance”? Please, remove it

Line 20: IC50 – 50 must be subscripted

Line 21: potent proliferative effect for what? Which cell line? Clarify it

Line 23: Delete “was selected for incorporation of the PRE”

Introduction

There are so many “Nowadays, currently, presently” occurrences in this section. Suggestion: Line 52: Remove “Currently”; Line 56: Remove “Presently”; Line 63: Remove “Recently”; Line 73: Remove “Nowadays”; Line 91: Remove Currently

Line 60: Remove Besides

Line 88: revise that statement. Suggestion: “Microemulsion systems are composed of oil, water, and surfactant, which feature….”

Line 93: Insert a reference for “gels”

Line 94: Which results are related to the increase of the solubility and absorption rate? Suggestion: Remove “Based on these results”

Line 99: Remove “Therefore”

Line 102: specify which “activities” were evaluated.

Line 113: Remove the “.” After 45

Line 129: how long the irradiation cycles were applied?

Line 148: Replace “as gradient condition” to “in gradient elution”

Line 200: Replace “their proliferation (cell proliferation assay)” to “in cell proliferation assays”

Line 231: Remove “.” After 0h

Line 222: Why the mitigation and the sircol collagen assays were not conducted with the PRE, the crude extract and the isolated compounds?

Line 243: insert a “the” after “The sample and”

Line 281: Is “The obtained extract” the PRE? Clarify it

Line 283: Why the off-white color of the obtained power can be classified as a satisfying physical appearance? Insert a reference or delete it.

Line 312: Revise “117.801.16%”

Lines 310-312: Is the decrease of the proliferative effect related to the solubility of PRE at higher concentrations? Or maybe cytotoxicity. Discuss these results.

Lines 314-315: If the increase in activity is related to the increase in the concentration of pentacyclic triterpenes, shouldn't the isolated compounds exhibit the highest percentages of proliferation? Could other factors be affecting this activity?

Line 336: Insert “(Table 3)” after the IC50 value, and remove it from the end of that paragraph.

Line 339: The cytotoxicity was evaluated in which cell line?

Line 343: Again, I do not believe that “a satisfying physical appearance” is relevant.

Lines 346-347: The PRE has antibacterial activity, and not the method – revise that sentence.

Line 352: concentrations

Figure 2: These photos do not provide relevant information as the transparency of the different microemulsions is not clear. Remove them and adequate the text.

Lines 411-413: revise that sentence

Figure 5: remove it

Line 453: Explain also the migration assay or delete that paragraph

Lines 456-470: The data is presented in Table 6. Discuss only the most relevant results.

Line 476: Insert PRE after 2.5%

Remove the “.” after “h” in all occurrences.

Lines 516-541: The results of the mitigation and proliferation assay show that the higher percentage of PRE does not result in increased activity. Did the authors consider conducting release assays to try to explain these results?

Figures 6 and 7: Remove them or put them in the SI

Lines 616-624: Do the authors consider these differences in collagen content relevant? Compare with other emulsions with the same application and discuss it.

Line 666: Remove “.” after hours

Lines 664-667: revise that sentence

Lines 682-683: Suggestion: Remove “To develop microemulsion for containing PRE”

Author Response

Reviewer 1

Comment 1

ABSTRACT – must be deeply revised

Line 18: the described method increased the pentacyclic triterpene content to 59.6% - that increase is concerning what? Revise it

Suggestion: “This method furnished a pentacyclic content of 59.6% w/w…”

Line 19: What do you mean by “satisfactory physical appearance”? Please, remove it

Line 20: IC50 – 50 must be subscripted

Line 21: potent proliferative effect for what? Which cell line? Clarify it

Line 23: Delete “was selected for incorporation of the PRE”

Response #1

Thank you for your comment. We re-checked and rewrote sentences in the revised version of the abstract, as shown below.

Revised version

A number of publications have shown that Centella asiatica (L.) Urb. and its active constituents (pentacyclic triterpenes) have an the effective on wound healing. The pentacyclic triterpenes-rich C. asiatica extract (PRE) was prepared following previous study by microwave-assisted extraction (MAE) and fractionation with macroporous resin. This method provided the pentacyclic triterpene content in the extract up to 59.60% w/w. The PRE showed potent anti-inflammatory activity by inhibiting nitric oxide (NO) production with an IC50 value of 20.59 ± 3.48 μg/mL and a potent fibroblast proliferative effect (165.67%) at concentrations of 10 μg/mL. The microemulsion consisted of water: oil: surfactant mixture of 2: 2: 6, using coconut oil: clove oil (1:1) as oil phase and Tween 20: Span 20 (2:1) as surfactant mixture and 1.0, 2.5, and 5.0% PRE were prepared. Cell proliferation, migration, and collagen production of microemulsion base and microemulsion containing 1.0, 2.5, and 5.0% PRE were evaluated. The results revealed that the microemulsion containing 1% PRE had the highest proliferation effect (136.30 ± 3.93 to 152.65 ± 3.48% at concentrations 10 μg/mL), migration activities (100.00 ± 0.0% at 24 h), and collagen production in human dermal fibroblast (HDF) and human gingival fibroblast (HGF) cells when compared with other formulations or blank. Moreover, the anti-inflammatory activity of microemulsions containing 1% PRE was slightly lower than standard indomethacin. Anti-inflammation of microemulsion containing PRE exhibited a dose-dependent trend, while 5% PRE was more potent than the standard drug. Considering the potent wound healing activities and the good anti-inflammatory activity of microemulsion containing PRE, the microemulsion with 1% PRE was identified as the most suitable oral spray formulation for oral ulcer treatment.

Comment 2

Introduction

There are so many “Nowadays, currently, presently” occurrences in this section. Suggestion: Line 52: Remove “Currently”; Line 56: Remove “Presently”; Line 63: Remove “Recently”; Line 73: Remove “Nowadays”; Line 91: Remove Currently

Line 60: Remove Besides

Line 88: revise that statement. Suggestion: “Microemulsion systems are composed of oil, water, and surfactant, which feature….”

Line 93: Insert a reference for “gels”

Line 94: Which results are related to the increase of the solubility and absorption rate? Suggestion: Remove “Based on these results”

Line 99: Remove “Therefore”

Line 102: specify which “activities” were evaluated.

Response #2

Thank you for your comment. We re-checked and rewrote sentences in the revision version of introduction following your recommendations.

Comment 3

Materials and Methods

Line 113: Remove the “.” After 45

Line 148: Replace “as gradient condition” to “in gradient elution”

Line 200: Replace “their proliferation (cell proliferation assay)” to “in cell proliferation assays”

Line 231: Remove “.” After 0h

Line 243: insert a “the” after “The sample and”

Response #3

Thank you for your comment. We re-checked and rewrote sentences in the revised version of Materials and methods.

Comment 4

Line 129: how long the irradiation cycles were applied?

Response #4

Thank you for your comment. We used 4 irradiation cycles, in which each cycle is 15 s power on and 30 s power off, as mentioned in the revised manuscript.

Comment 5

             Line 222: Why the mitigation and the sircol collagen assays were not conducted with the PRE, the crude extract and the isolated compounds?

Response #5

            Thank you for your comment. The biological analysis (cell proliferation assay and anti-inflammation) of PRE, crude extract of C. asiatica, and pure isolated compounds (MS, AS, MA, and AA) were conducted. The comparison of wound healing capacities represents the rationale for selecting PRE as the active compound in the formulation. Cell proliferation, anti-inflammation, collagen production, and cell migration properties of the microemulsion base and the formulation were compared. The efficacy results indicated that samples containing PRE was better than pure base alone and is suitable for oral ulcer treatment preparation.

Comment 6

Results and Discussion

Line 281: Is “The obtained extract” the PRE? Clarify it

Line 283: Why the off-white color of the obtained power can be classified as a satisfying physical appearance? Insert a reference or delete it.

Line 312: Revise “117.801.16%”

Line 336: Insert “(Table 3)” after the IC50 value, and remove it from the end of that paragraph.

Line 339: The cytotoxicity was evaluated in which cell line?

Line 343: Again, I do not believe that “a satisfying physical appearance” is relevant.

Lines 346-347: The PRE has antibacterial activity, and not the method – revise that sentence.

Line 352: concentrations

Lines 411-413: revise that sentence

Figure 5: remove it

Line 453: Explain also the migration assay or delete that paragraph

Line 476: Insert PRE after 2.5%

Remove the “.” after “h” in all occurrences.

Figures 6 and 7: Remove them or put them in the SI

Line 666: Remove “.” after hours

Lines 664-667: revise that sentence.

Lines 682-683: Suggestion: Remove “To develop microemulsion for containing PRE”

 Response #6

Thank you for your comment. We re-checked, removed some pictures or words, and rewrote the sentences in the revised version of results and discussion section as recommended.

Comment 7

Lines 310-312: Is the decrease of the proliferative effect related to the solubility of PRE at higher concentrations? Or maybe cytotoxicity. Discuss these results.

Lines 314-315: If the increase in activity is related to the increase in the concentration of pentacyclic triterpenes, shouldn't the isolated compounds exhibit the highest percentages of proliferation? Could other factors be affecting this activity?

 Response #7

            Thank you for your suggestion. We discussed and rewrote the sentence for greater clarity, as shown below.

Original version

The cell viability and proliferation of all tested compounds at 1, 3, 10, 30, and 100 μg/mL are presented in Table 2. It was found that the enhancement effect on HDF cell proliferation was achieved by the MTT method. The results showed that PRE exhibited a potent proliferative effect (165.67%) at concentrations of 10 μg/mL, while the proliferative effect was decreased when using higher concentrations at 30 and 100 μg/mL (148.51±1.96% and 117.801.16%), respectively. The PRE from C. asiatica showed a higher proliferation property than the crude extract and pure isolated compounds (MS, AS, MA, and AA) due to increasing the pentacyclic triterpene content in the PRE. Compared to crude extract, higher content of pentacyclic triterpenes, better biological activities, better dose adjustment, and better physical appearance were observed in PRE. Moreover, our finding, corresponding with previous reports on the chemical complexity of the extract, has a greater advantage than pure isolated compounds in terms of the wide range of pharmacological activities and is also cheaper [19].

Revised version

The cell viability and proliferation of all tested compounds at 1, 3, 10, 30, and 100 μg/mL are presented in Table 2. It was found that the enhancement effect on HDF cell proliferation was achieved by the MTT method. The results showed that PRE exhibited a potent proliferative effect (165.67 ± 1.95%) at concentrations of 10 μg/mL and decreased when using higher concentrations at 30 and 100 μg/mL (148.51 ± 1.96% and 117.80 ± 1.16%), respectively. These results may be due to the cytotoxicity, which was also observed in the previous study that high doses of pentacyclic triterpenes could decrease cell viability [43]. The PRE showed a higher proliferation property than the crude extract and pure isolated compounds (MS, AS, MA, and AA) due to the increase in pentacyclic triterpene content in the PRE. Higher content of pentacyclic triterpenes, better biological activities, dose adjustment, and physical appearance were observed in PRE when compared to the crude extract. Our findings, in-line with previous reports on the chemical complexity of the extract, has a greater advantage over pure isolated compounds in terms of the wide range of pharmacological activities, lower toxicity, and is also cost effective [19].

Comment 8

             Figure 2: These photos do not provide relevant information as the transparency of the different microemulsions is not clear. Remove them and adequate the text.

Response #8

            Thank you for your suggestion. We designed to move 4 figures to the supplement data.

Comment 9

             Lines 516-541: The results of the mitigation and proliferation assay show that the higher percentage of PRE does not result in increased activity. Did the authors consider conducting release assays to try to explain these results?

Response #9

Thank you for your suggestion, we did not conduct the release study to explain the phenomenon. We think this effect was due to the higher concentration of PRE or high doses of pentacyclic triterpenes could decrease cell viability as same as the results from the previous study [34].

Comment 10

             Lines 456-470: The data is presented in Table 6. Discuss only the most relevant results.

Response #10

            Thank you for your suggestion. We rewrote sentences in the revised version, as shown below.

Original version

It was found that the enhancement effect on HDF cell proliferation was achieved by the MTT method. The results showed that microemulsions containing 1% and 2.5% PRE exhibited a potent proliferative effect (123.91-136.30%) at 3 and 10 μg/mL. While microemulsion containing 5% PRE acted as a proliferative inducer (108.54 ± 4.67%) at low concentrations (1 μg/mL), and microemulsion base showed the lowest percent proliferation (<100%) on HDF cells. The microemulsion containing 1% PRE showed the highest proliferation property at concentrations 10 μg/mL (136.30 ± 3.93%), 2.5% PRE showed the highest proliferation property at concentrations 3 μg/mL (123.91 ± 3.96%), and 5% PRE showed the highest proliferation property at concentrations 1 μg/mL (108.54 ± 4.67%). Each sample acted as a low proliferative at higher concentrations. Our results suggested that pentacyclic triterpenes content in each formulation plays an important role in cell proliferation. It is consistent with previous test results on cell proliferation of PRE that showed the best activity at the optimum concentration. The increasing cell viability of each sample at 0.3, 1, 3, 10, and 30 μg/mL are presented in Table 6. The experiment supported that the microemulsions containing PRE stimulate the growth of HDF cells.

Revised version

It was found that the enhancement effect on HDF cell proliferation was achieved by the MTT method. The results showed that microemulsions containing 1% and 2.5% PRE exhibited a potent proliferative effect (123.91-136.30%) at 3 and 10 μg/mL. While microemulsion containing 5% PRE acted as a proliferative inducer (108.54 ± 4.67%) at low concentrations (1 μg/mL), and microemulsion base showed the lowest percent proliferation (<100%) on HDF cells. The microemulsion containing 1%, 2.5%, and 5% PRE showed the highest proliferation property at concentrations of 10, 3, and 1 μg/mL, respectively. All samples acted as a low proliferative at higher concentrations. Our results suggested that pentacyclic triterpenes content in each formulation plays an important role in cell proliferation. It is consistent with previous test results on cell proliferation of PRE that showed the best activity at the optimum concentration [43, 44]. The increasing cell viability of each sample at 0.3, 1, 3, 10, and 30 μg/mL are presented in Table 6. It should be noted that all microemulsions containing PRE stimulate the growth of HDF cells.

Comment 11

             Lines 616-624: Do the authors consider these differences in collagen content relevant? Compare with other emulsions with the same application and discuss it.

Response #11

Thank you for your suggestion. We rewrote sentences in the revised version, as shown below.

Original version

Numerous studies have documented the action of triterpenes from C. asiatica, such as MS, MA, and AA, on the synthesis of collagen related to the wound-healing process [37]. The experiments of the Sircol collagen assay provide the information that the substances can be added to the microemulsion formula to stimulate collagen production. The collagen content in the medium of fibroblasts incubated with the test sample was measured. Collagen production on HDF and HGF cells by microemulsion base and microemulsion containing PRE at various concentrations were evaluated and compared with the control. For HDF cells, it was found that the microemulsion containing 1% PRE exhibited the highest collagen content of 55.80 ± 0.54 μg/mL followed by base (53.80 ± 1.15 μg/mL), 2.5% PRE (53.10 ± 1.91 μg/mL) and 5% PRE (51.50 ± 0.94), respectively. For HGF cells, it was found that the collagen content of microemulsion containing 1% PRE also showed the highest collagen content at 52.80 ± 0.54 μg/mL followed by 5% PRE (50.10 ± 0.94), 2.5% PRE (49.60 ± 1.91) and base (46.80 ± 0.54), respectively. From the result mentioned above, after the statistical test, it could be concluded that all formulations tested revealed slightly significant differences better than the control (p<0.05). However, all formulations tested did not show any significant difference between each formulation. The results obtained are shown in Table 10:

Revised version

Numerous studies have documented the action of triterpenes from C. asiatica, such as MS, MA, and AA, on the synthesis of collagen related to the wound-healing process [46,47,48]. The experiments of the Sircol collagen assay provide the information that the substances can be added to the microemulsion formula to stimulate collagen production. The collagen content in the medium of fibroblasts incubated with the test sample was measured. Collagen production on HDF and HGF cells by microemulsion base and microemulsion containing PRE at various concentrations were evaluated and compared with the control. For HDF cells, it was found that the microemulsion containing 1% PRE exhibited the highest collagen content of 55.80 ± 0.54 μg/mL followed by base (53.80 ± 1.15 μg/mL), 2.5% PRE (53.10 ± 1.91 μg/mL) and 5% PRE (51.50 ± 0.94), respectively. For HGF cells, it was found that the collagen content of microemulsion containing 1% PRE also showed the highest collagen content at 52.80 ± 0.54 μg/mL followed by 5% PRE (50.10 ± 0.94 μg/mL), 2.5% PRE (49.60 ± 1.91 μg/mL) and base (46.80 ± 0.54 μg/mL), respectively. Combining with statistical analysis, it could be concluded that all formulations tested revealed slightly significant differences better than the control (p<0.05). However, all formulations tested did not show a significant difference between each formulation. Collagen production in both cells was not different between base and various concentrations. This may be due to the coconut oil in the preparation, which could also promote collagen production [49,50]. The results revealed that the combination of PRE and coconut oil facilitated collagen production in fibroblast cells. The results obtained are shown in Table 10. 

Reviewer 2 Report

Recommendations :

1. Specify the control results in Table 8.

2. Include more scientific sources in the introduction of the manuscript and discussion of results.

Author Response

Reviewer 2

Comment 1

Specify the control results in Table 8.

Response #1

Thank you for your comment. We removed the specific control results in the revision version.

Comment 2

Include more scientific sources in the introduction of the manuscript and discussion of results.

Response #2

             Thanks for your comments. We rechecked and re-written the introduction of the manuscript and discussion of results for more understanding and add more discussion as reviewers suggested.

Reviewer 3 Report

The manuscript entitled " Development of oral microemulsion spray containing pentacyclic triterpenes-rich C. asiatica extract for healing the mouth ulcers" presented by Sanguansajapong et al, summaries a research study towards role of pentacyclic triterpenes fraction having mouth healing ulcers procedure. Overall, the manuscript is addressing and delivering the scientific content. However, some major flaws need to be addressed for further improvement:   

·       Title must not contain any abbreviation eg C. asiatica

·       Why there is need of microemulsion is there any conventional formulation already existing or any comparative study where bioavailability or permeability issue solved. Author need to mention the reason behind selection of the formulation in context of pentacyclic triterpenes of C. asiatica. Please explain in introduction

·       Under material and method why plant is heated with 60° C, why shade dried or less temperature not tried…as it contains lot of thermolabile compounds

·       Table 4.1 is missing…line 282 Page no 6

·       Where is result of HPLC quantification for marker compounds. How pentacyclic rich fraction was estimated. Is there any chemical test or total terpenoids content was estimated?

·       Please discuss your result with reference to one earlier study published on same https://pubmed.ncbi.nlm.nih.gov/10356407/

·       Language and any other typological mistake can be address

Author Response

Reviewer 3

Comment 1

Title must not contain any abbreviation eg C. asiatica

Response #1

            Thank you for your recommendation. We changed C. asiatica to Centella asiatica L. Urb. (full name) the specific control results in the revision version, as shown below.

Original version

Development of oral microemulsion spray containing pentacyclic triterpenes-rich C. asiatica extract for healing the mouth ulcers

Revised version

Development of oral microemulsion spray containing pentacyclic triterpenes-rich Centella asiatica (L.) Urb. extract for healing mouth ulcers

Comment 2

Why there is need of microemulsion is there any conventional formulation already existing or any comparative study where bioavailability or permeability issue solved. Author need to mention the reason behind selection of the formulation in context of pentacyclic triterpenes of C. asiatica. Please explain in introduction

Response #2

Thank you for your comment. We have added the reason behind the selection of microemulsion in the introduction part as follows.

According to our preliminary study, the limited amount of the pentacyclic triterpenes-rich C. asiatica extract (PRE) (0.1% w/w) could be incorporated into the aqueous solution. However, higher concentrations of the extract incorporated caused the precipitation of the undissolved moieties. PRE consist of  asiatic acid, and madecassic acid, which are highly lipophilic compounds, and asiaticoside, madecassoside which are hydrophilic compounds. These compounds were aimed to codeliver to the target site to exert their activities. For these reasons, the development of formulations that provide the high solubilizing effect of PRE and could carry the extract to the oral mucosa efficiently are of interest where microemulsion system was developed to address these problems.                      

Comment 3

Under material and method why plant is heated with 60° C, why shade dried or less temperature not tried…as it contains lot of thermolabile compounds.

Response #3

            Thank you for your comment. We prepared PRE, followed by a previous study [17]. Moreover, fresh C. asiatic contains a lot of water, and Thailand has high humidity. Shade-dried or less temperature could take a very long time and increase the risk of microbial growth and contamination.

Comment 4

  • Table 4.1 is missing…line 282 Page no 6

Response #4

Original version

the increase in the pentacyclic triterpene content up to 59.60% w/w (Table 4.1)

Revised version

             the increase in the pentacyclic triterpene content up to 59.60% w/w (Table 1)

Comment 5

Where is result of HPLC quantification for marker compounds. How pentacyclic rich fraction was estimated. Is there any chemical test or total terpenoids content was estimated?

Response #5

            Thank you for your comment. We explained more in section 2.4. The HPLC quantitative analysis is followed by a previous study as mentioned in the material and method. The pentacyclic triterpenes content in each sample was compared the area under the curve with the standard curve of the standards of madecassoside (MS), asiaticoside (AS), madecassic acid (MA), and asiatic acid (AA) at the concentration of between 0.03 - 0.50 μg/mL. The concentration of pentacyclic triterpenes content in the extract is the sum of the amount of the 4 standard compounds, which are the major pentacyclic triterpenes in C. asiatica, in the extract. Even though there are some other pentacyclic triterpenes in the extract, we used 4 major compounds to quantify the pentacyclic triterpene content not to be lower than these amounts. The content of pentacyclic triterpenes are present in Table 1.

Comment 6

Please discuss your result with reference to one earlier study published on same https://pubmed.ncbi.nlm.nih.gov/10356407/

Response #6

Thank you for your recommendation. We added the detail of the comparison with the     titrated extract of Centella asiatica (TECA) and ECa233 in section 3.1.1.

Comment 7

Language and any other typological mistake can be address

Response #7

             Thank you for your suggestion, we have checked the typological mistakes and sent this manuscript to be edited by the native English speaker (Mrs. Carissa Yvonne Sirikitputtisak).

Reviewer 4 Report

The authors worked on “Development of oral microemulsion spray containing pentacy-cyclic triterpenes-rich C. asiatica extract for healing the mouth ulcers”. The original research article is very well written and justified through suitable evaluation parameters and references. Though it contains sufficient results to be accepted for publication, but still minor modifications are recommended to improve the quality of the manuscript.

Reviewer comments and suggestions;

1. The whole manuscript should be revised for grammatical and typographic mistakes.  

2. There should be a space between the digit and unit as in lines 191 and 195, add the spaces between the digit and unit. Same is for the rest of the manuscript.

3. The reagents used for the preparation of microemulsion should be explained with sufficient literature in the introduction section.

4. The problems should be addressed in the introduction section while novelty should be explained at the last paragraph of the introduction for better understanding of the reader.  

Conclusion

Overall, the manuscript could be considered as scientific rigor and seems able to add in existing scientific knowledge. Therefore, I recommend the Acceptance of manuscript with minor modifications on above mentioned comments.

The authors worked on “Development of oral microemulsion spray containing pentacy-cyclic triterpenes-rich C. asiatica extract for healing the mouth ulcers”. The original research article is very well written and justified through suitable evaluation parameters and references. Though it contains sufficient results to be accepted for publication, but still minor modifications are recommended to improve the quality of the manuscript.

Reviewer comments and suggestions;

1. The whole manuscript should be revised for grammatical and typographic mistakes.  

2. There should be a space between the digit and unit as in lines 191 and 195, add the spaces between the digit and unit. Same is for the rest of the manuscript.

3. The reagents used for the preparation of microemulsion should be explained with sufficient literature in the introduction section.

4. The problems should be addressed in the introduction section while novelty should be explained at the last paragraph of the introduction for better understanding of the reader.  

Conclusion

Overall, the manuscript could be considered as scientific rigor and seems able to add in existing scientific knowledge. Therefore, I recommend the Acceptance of manuscript with minor modifications on above mentioned comments.

The authors worked on “Development of oral microemulsion spray containing pentacy-cyclic triterpenes-rich C. asiatica extract for healing the mouth ulcers”. The original research article is very well written and justified through suitable evaluation parameters and references. Though it contains sufficient results to be accepted for publication, but still minor modifications are recommended to improve the quality of the manuscript.

Reviewer comments and suggestions;

1. The whole manuscript should be revised for grammatical and typographic mistakes.  

2. There should be a space between the digit and unit as in lines 191 and 195, add the spaces between the digit and unit. Same is for the rest of the manuscript.

3. The reagents used for the preparation of microemulsion should be explained with sufficient literature in the introduction section.

4. The problems should be addressed in the introduction section while novelty should be explained at the last paragraph of the introduction for better understanding of the reader.  

Conclusion

Overall, the manuscript could be considered as scientific rigor and seems able to add in existing scientific knowledge. Therefore, I recommend the Acceptance of manuscript with minor modifications on above mentioned comments.

The authors worked on “Development of oral microemulsion spray containing pentacy-cyclic triterpenes-rich C. asiatica extract for healing the mouth ulcers”. The original research article is very well written and justified through suitable evaluation parameters and references. Though it contains sufficient results to be accepted for publication, but still minor modifications are recommended to improve the quality of the manuscript.

Reviewer comments and suggestions;

1. The whole manuscript should be revised for grammatical and typographic mistakes.  

2. There should be a space between the digit and unit as in lines 191 and 195, add the spaces between the digit and unit. Same is for the rest of the manuscript.

3. The reagents used for the preparation of microemulsion should be explained with sufficient literature in the introduction section.

4. The problems should be addressed in the introduction section while novelty should be explained at the last paragraph of the introduction for better understanding of the reader.  

Conclusion

Overall, the manuscript could be considered as scientific rigor and seems able to add in existing scientific knowledge. Therefore, I recommend the Acceptance of manuscript with minor modifications on above mentioned comments.

The authors worked on “Development of oral microemulsion spray containing pentacy-cyclic triterpenes-rich C. asiatica extract for healing the mouth ulcers”. The original research article is very well written and justified through suitable evaluation parameters and references. Though it contains sufficient results to be accepted for publication, but still minor modifications are recommended to improve the quality of the manuscript.

Reviewer comments and suggestions;

1. The whole manuscript should be revised for grammatical and typographic mistakes.  

2. There should be a space between the digit and unit as in lines 191 and 195, add the spaces between the digit and unit. Same is for the rest of the manuscript.

3. The reagents used for the preparation of microemulsion should be explained with sufficient literature in the introduction section.

4. The problems should be addressed in the introduction section while novelty should be explained at the last paragraph of the introduction for better understanding of the reader.  

Conclusion

Overall, the manuscript could be considered as scientific rigor and seems able to add in existing scientific knowledge. Therefore, I recommend the Acceptance of manuscript with minor modifications on above mentioned comments.

The authors worked on “Development of oral microemulsion spray containing pentacy-cyclic triterpenes-rich C. asiatica extract for healing the mouth ulcers”. The original research article is very well written and justified through suitable evaluation parameters and references. Though it contains sufficient results to be accepted for publication, but still minor modifications are recommended to improve the quality of the manuscript.

Reviewer comments and suggestions;

1. The whole manuscript should be revised for grammatical and typographic mistakes.  

2. There should be a space between the digit and unit as in lines 191 and 195, add the spaces between the digit and unit. Same is for the rest of the manuscript.

3. The reagents used for the preparation of microemulsion should be explained with sufficient literature in the introduction section.

4. The problems should be addressed in the introduction section while novelty should be explained at the last paragraph of the introduction for better understanding of the reader.  

Conclusion

Overall, the manuscript could be considered as scientific rigor and seems able to add in existing scientific knowledge. Therefore, I recommend the Acceptance of manuscript with minor modifications on above mentioned comments.

The authors worked on “Development of oral microemulsion spray containing pentacy-cyclic triterpenes-rich C. asiatica extract for healing the mouth ulcers”. The original research article is very well written and justified through suitable evaluation parameters and references. Though it contains sufficient results to be accepted for publication, but still minor modifications are recommended to improve the quality of the manuscript.

Reviewer comments and suggestions;

1. The whole manuscript should be revised for grammatical and typographic mistakes.  

2. There should be a space between the digit and unit as in lines 191 and 195, add the spaces between the digit and unit. Same is for the rest of the manuscript.

3. The reagents used for the preparation of microemulsion should be explained with sufficient literature in the introduction section.

4. The problems should be addressed in the introduction section while novelty should be explained at the last paragraph of the introduction for better understanding of the reader.  

Conclusion

Overall, the manuscript could be considered as scientific rigor and seems able to add in existing scientific knowledge. Therefore, I recommend the Acceptance of manuscript with minor modifications on above mentioned comments.

Author Response

Reviewer 4

Comment 1

The whole manuscript should be revised for grammatical and typographic mistakes. 

Response #1

        Thank you for your suggestion, we have checked the typological mistakes and sent this manuscript to be edited by the native English speaker.

Comment 2

There should be a space between the digit and unit as in lines 191 and 195, add the spaces between the digit and unit. Same is for the rest of the manuscript

Response #2

Thank you for your comment. We have checked and edited throughout the manuscript already.

Comment 3

The reagents used for the preparation of microemulsion should be explained with sufficient literature in the introduction section.

Response #3

             We have added the short review on the excipients used in the microemulsion formulations in the introduction section and added more references.

Comment 4

The problems should be addressed in the introduction section while novelty should be explained at the last paragraph of the introduction for better understanding of the reader. 

Response #4

Thank you for your comment. We have added the reasons behind the selection of microemulsion in the introduction part as follows

According to our preliminary study, the limited amount of the pentacyclic triterpenes-rich C. asiatica extract (PRE) (0.1% w/w) could be incorporated into the aqueous solution. However, higher concentrations of the extract incorporated caused the precipitation of the undissolved moieties. PRE consist of asiatic acid, and madecassic acid, which are highly lipophilic compounds, and asiaticoside, madecassoside which are hydrophilic compounds. These compounds were aimed to codeliver to the target site to exert their activities. For these reasons, the development of formulations that provide the high solubilizing effect of PRE and could carry the extract to the oral mucosa efficiently are of interest where microemulsion system was developed to address these problems.

Reviewer 5 Report

Overview and general recommendation:

The study was focused on developing microemulsions containing pentacyclic-triterpens from Centella asiatica extract intended for healing of mouth ulcers. The abstract is appropriate for the content of the text. The article is well constructed and clearly written and the experiments were well conducted by generally sound methodology. The methods are clearly described, and the results are supported by enough references and statistical analysis.

            However, the authors should clarify the following:

-          In the title of the manuscript the authors claim to develop <<… oral microemulsion spray…>>>, and in section 5. Conclusions, lines 693-694 they state that <<<< This study achieved the objectives of formulating an oral spray for healing oral ulcers. >>> but all the presented determinations are conducted on microemulsion systems, not on the final product.

-          The authors indicate the use of clove oil in the subchapter 2.5. Preparation of pseudo-ternary phase diagram of microemulsion, but they don’t referred to this essential oil in the section 2.2. Materials and Methods and neither do they present the reasons for its selection as part of the oil phase of the microemulsions.

-          The authors should present more explicit the control used for the Cell viability and proliferation assay and also for the Migration assay.  

-          In the section 3. Results and Discussion, at subsection 3.1.1. Preparation of C. asiatica crude extracts and PRE at lines 283-284 and also in the section 5. Conclusions at line 680 the authors describe the PRE as a white powder, but at subsection 3.2.1. Formulation of microemulsion containing PRE at lines 419-420 the microemulsions containing PRE present a <<<…light yellowish to brown color…>>> depending on the used amount of PRE. Please explain.

-          In the subsection 3.2. Evaluation of microemulsion base and microemulsion containing PRE in various concentration, at Figure 3. Pseudo-ternary phase diagram M1-M6 – the last two diagrams M5 and M6 present the same microemulsion components, but different microemulsion area.

Also minor typos are present, like:

-          Line 195: <<<…with an angle of detection at 90ºC. >>>

Author Response

Reviewer 5

Comment 1

In the title of the manuscript the authors claim to develop <<… oral microemulsion spray…>>>, and in section 5. Conclusions, lines 693-694 they state that <<<< This study achieved the objectives of formulating an oral spray for healing oral ulcers. >>> but all the presented determinations are conducted on microemulsion systems, not on the final product. 

Response #1

            This study aimed to develop the oral spray using the microemulsion containing PRE. The physical properties of this microemulsion system were evaluated to select the most suitable formulation for an oral spray. We did not add more excipients, e.g., film former, in this system and used this microemulsion as a final product in this study.

Comment 2

The authors indicate the use of clove oil in the subchapter 2.5. Preparation of pseudo-ternary phase diagram of microemulsion, but they don’t referred to this essential oil in the section 2.2. Materials and Methods and neither do they present the reasons for its selection as part of the oil phase of the microemulsions.

Response #2

Thank you for your comment. We have added the clove oil and PG information in section 2.2. In addition, we have added the reason of selecting clove oil and other microemulsion excipients in the introduction part as follows;

In this study, the microemulsion systems were prepared using coconut oil or the combination of coconut oil and clove oil as an oil phase. Coconut oil is an edible oil which has been used in different emulsion systems e.g. micro- and nanoemulsion [30,31,32]. It possesses antibacterial, anti-inflammatory, antioxidant, and wound healing properties on skin and oral mucosa [33,34]. Clove oil has been utilized in toothache and oral pain relief. It also exhibits antimicrobial, antioxidant and anti-inflammatory effects [35,36]. These ingredients were aimed to aid in the oral ulcer treatment.  Tween 20 and Span 20 were chosen as surfactants due to their non-ionic features and safety for use in oral products [37]. Propylene glycol (PG) was occasionally used as a cosurfactant to stabilize the microemulsion droplets [38].

This research aimed to develop the oral microemulsion spray containing PRE for  healing mouth ulcers. The oral ulcer healing potential of the PRE were investigated and incorporated into the  microemulsion system. The obtained microemulsion was then evaluated for its activities, including wound healing activity on human dermal fibroblast (HDF) and human gingival fibroblast (HGF) and anti-inflammation by inhibition of nitric oxide (NO) production from macrophage cells to ensure its potential for oral ulcer treatment. The results of this study could be used for new herbal-based microemulsion product development for oral ulcer treatment.

Comment 3

The authors should present more explicit the control used for the Cell viability and proliferation assay and also for the Migration assay

Response #3

             Thank you for your comments. We added the detail of the control in the cell viability, proliferation assay, and migration assay.

Comment 4

In the section 3. Results and Discussion, at subsection 3.1.1. Preparation of C. asiatica crude extracts and PRE at lines 283-284 and also in the section 5. Conclusions at line 680 the authors describe the PRE as a white powder, but at subsection 3.2.1. Formulation of microemulsion containing PRE at lines 419-420 the microemulsions containing PRE present a <<<…light yellowish to brown color…>>> depending on the used amount of PRE. Please explain.

Response #4

Thank you for your comment. From our results, the PRE was an off-white powder, however as it dissolved in the microemulsion system, it became a light yellowish to brown color depending on the concentration of PRE added.

Comment 5

In the subsection 3.2. Evaluation of microemulsion base and microemulsion containing PRE in various concentration, at Figure 3. Pseudo-ternary phase diagram M1-M6 – the last two diagrams M5 and M6 present the same microemulsion components, but different microemulsion area.

Response #5

Thank you for your comment, and sorry for our mistake. We have edited the caption of the M6 diagram in the figure file in which the oil phase should be coconut oil: clove oil = 2:1.

Comment 6

-Also minor typos are present, like:    Line 195: <<<…with an angle of detection at 90ºC. >>>

Response #6

             We have edited   90ºC to 90º in the section 2.7.6.

Round 2

Reviewer 1 Report

The authors worked properly to review the manuscript. However, some points in the introduction still deserve attention. In the attached file the authors can find some errors, but I suggest a thorough review by the authors.

In upcoming manuscript review processes, I suggest reviewers use Word's “Track Changes” tool. The markings were very confusing as they were made.

I also suggest that the authors answer the reviewers' questions point by point, and not in blocks as was done.

Author Response

Response to reviewer comments of the manuscript entitled "Developing microemulsions containing pentacyclic-triterpens from Centella asiatica extract intended for healing of mouth ulcers"

Reviewer 1

Comment 1

ABSTRACT – must be deeply revised

Line 18: the described method increased the pentacyclic triterpene content to 59.6% - that increase is concerning what? Revise it

Suggestion: “This method furnished a pentacyclic content of 59.6% w/w…”

Line 19: What do you mean by “satisfactory physical appearance”? Please, remove it

Line 20: IC50 – 50 must be subscripted

Line 21: potent proliferative effect for what? Which cell line? Clarify it

Line 23: Delete “was selected for incorporation of the PRE”

Response #1

Thank you for your comment. We re-checked and rewrote sentences in the revised version of the abstract, as shown below.

Revised version

Abstract: Several publications have shown that Centella asiatica (L.) Urb. and its active constituents (pentacyclic triterpenes) are effective in wound healing. The pentacyclic triterpenes-rich C. asiatica extract (PRE) was prepared following a previous study by microwave-assisted extraction (MAE) and fractionation with macroporous resin. This method provided the pentacyclic triterpene content in the extract up to 59.60% w/w. The PRE showed potent anti-inflammatory activity by inhibiting nitric oxide (NO) production with an IC50 value of 20.59 ± 3.48 μg/mL and a potent fibroblast proliferative effect (165.67%) at concentrations of 10 μg/mL. The prepared microemulsion consisted of water: oil: surfactant mixture of 2: 2: 6, using coconut oil: clove oil (1:1) as oil phase and Tween 20: Span 20 (2:1) as surfactant mixture and 1.0, 2.5, and 5.0% PRE. Cell proliferation, migration, and collagen production of microemulsion base and microemulsion containing 1.0, 2.5, and 5.0% PRE were evaluated. The results revealed that the microemulsion containing 1% PRE had the highest proliferation effect (136.30 ± 3.93 to 152.65 ± 3.48% at concentrations 10 μg/mL), migration activities (100.00 ± 0.0% at 24 h), and collagen production in human dermal fibroblast (HDF) and human gingival fibroblast (HGF) cells when compared with other formulations or blank. Moreover, the anti-inflammatory activity of microemulsions containing 1% PRE was slightly lower than standard indomethacin. Anti-inflammation of microemulsion containing PRE exhibited a dose-dependent trend, while 5% PRE was more potent than the standard drug. Considering the potent wound healing activities and the good anti-inflammatory activity of microemulsion containing PRE, the microemulsion with 1% PRE was identified as the most suitable oral spray formulation for oral ulcer treatment.

Comment 2

Introduction

There are so many “Nowadays, currently, presently” occurrences in this section. Suggestion: Line 52: Remove “Currently”; Line 56: Remove “Presently”; Line 63: Remove “Recently”; Line 73: Remove “Nowadays”; Line 91: Remove Currently

Line 60: Remove Besides

Line 88: revise that statement. Suggestion: “Microemulsion systems are composed of oil, water, and surfactant, which feature….”

Line 93: Insert a reference for “gels”

Line 94: Which results are related to the increase of the solubility and absorption rate? Suggestion: Remove “Based on these results”

Line 99: Remove “Therefore”

Line 102: specify which “activities” were evaluated.

Response #2

Thank you for your comment. We re-checked and rewrote sentences in the revision version of introduction following your recommendations.

Comment 3

Materials and Methods

Line 113: Remove the “.” After 45

Line 148: Replace “as gradient condition” to “in gradient elution”

Line 200: Replace “their proliferation (cell proliferation assay)” to “in cell proliferation assays”

Line 231: Remove “.” After 0h

Line 243: insert a “the” after “The sample and”

Response #3

Thank you for your comment. We re-checked and rewrote sentences in the revised version of Materials and methods.

Comment 4

Line 129: how long the irradiation cycles were applied?

Response #4

Thank you for your comment. We used 4 irradiation cycles, in which each cycle is 15 s power on and 30 s power off, as mentioned in the revised manuscript.

Comment 5

             Line 222: Why the mitigation and the sircol collagen assays were not conducted with the PRE, the crude extract and the isolated compounds?

Response #5

            Thank you for your comment. The biological analysis (cell proliferation assay and anti-inflammation) of PRE, crude extract of C. asiatica, and pure isolated compounds (MS, AS, MA, and AA) were conducted. The comparison of wound healing capacities represents the rationale for selecting PRE as the active compound in the formulation. Cell proliferation, anti-inflammation, collagen production, and cell migration properties of the microemulsion base and the formulation were compared. The efficacy results indicated that samples containing PRE was better than pure base alone and is suitable for oral ulcer treatment preparation.

Comment 6

Results and Discussion

Line 281: Is “The obtained extract” the PRE? Clarify it

Line 283: Why the off-white color of the obtained power can be classified as a satisfying physical appearance? Insert a reference or delete it.

Line 312: Revise “117.801.16%”

Line 336: Insert “(Table 3)” after the IC50 value, and remove it from the end of that paragraph.

Line 339: The cytotoxicity was evaluated in which cell line?

Line 343: Again, I do not believe that “a satisfying physical appearance” is relevant.

Lines 346-347: The PRE has antibacterial activity, and not the method – revise that sentence.

Line 352: concentrations

Lines 411-413: revise that sentence

Figure 5: remove it

Line 453: Explain also the migration assay or delete that paragraph

Line 476: Insert PRE after 2.5%

Remove the “.” after “h” in all occurrences.

Figures 6 and 7: Remove them or put them in the SI

Line 666: Remove “.” after hours

Lines 664-667: revise that sentence.

Lines 682-683: Suggestion: Remove “To develop microemulsion for containing PRE”

 Response #6

Thank you for your comment. We re-checked, removed some pictures or words, and rewrote the sentences in the revised version of results and discussion section as recommended.

Comment 7

Lines 310-312: Is the decrease of the proliferative effect related to the solubility of PRE at higher concentrations? Or maybe cytotoxicity. Discuss these results.

Lines 314-315: If the increase in activity is related to the increase in the concentration of pentacyclic triterpenes, shouldn't the isolated compounds exhibit the highest percentages of proliferation? Could other factors be affecting this activity?

 Response #7

            Thank you for your suggestion. We discussed and rewrote the sentence for greater clarity, as shown below.

Original version

The cell viability and proliferation of all tested compounds at 1, 3, 10, 30, and 100 μg/mL are presented in Table 2. It was found that the enhancement effect on HDF cell proliferation was achieved by the MTT method. The results showed that PRE exhibited a potent proliferative effect (165.67%) at concentrations of 10 μg/mL, while the proliferative effect was decreased when using higher concentrations at 30 and 100 μg/mL (148.51±1.96% and 117.801.16%), respectively. The PRE from C. asiatica showed a higher proliferation property than the crude extract and pure isolated compounds (MS, AS, MA, and AA) due to increasing the pentacyclic triterpene content in the PRE. Compared to crude extract, higher content of pentacyclic triterpenes, better biological activities, better dose adjustment, and better physical appearance were observed in PRE. Moreover, our finding, corresponding with previous reports on the chemical complexity of the extract, has a greater advantage than pure isolated compounds in terms of the wide range of pharmacological activities and is also cheaper [19].

Revised version

The cell viability and proliferation of all tested compounds at 1, 3, 10, 30, and 100 μg/mL are presented in Table 2. It was found that the enhancement effect on HDF cell proliferation was achieved by the MTT method. The results showed that PRE exhibited a potent proliferative effect (165.67 ± 1.95%) at 10 μg/mL and decreased when using higher concentrations at 30 and 100 μg/mL (148.51 ± 1.96% and 117.80 ± 1.16%), respectively. These results may be due to the cytotoxicity, which was also observed in the previous study that high doses of pentacyclic triterpenes could decrease cell viability [43]. The PRE showed a higher proliferation property than the crude extract and pure isolated compounds (MS, AS, MA, and AA) due to the increase in pentacyclic triterpene content in the PRE. Higher content of pentacyclic triterpenes, better biological activities, dose adjustment, and physical appearance were observed in PRE when compared to the crude extract. Our findings, in-line with previous reports on the chemical complexity of the extract, has a greater advantage over pure isolated compounds in terms of the wide range of pharmacological activities, lower toxicity, and is also cost effective [19].

Comment 8

             Figure 2: These photos do not provide relevant information as the transparency of the different microemulsions is not clear. Remove them and adequate the text.

Response #8

            Thank you for your suggestion. We designed to move 4 figures to the supplement data.

Comment 9

             Lines 516-541: The results of the mitigation and proliferation assay show that the higher percentage of PRE does not result in increased activity. Did the authors consider conducting release assays to try to explain these results?

Response #9

Thank you for your suggestion, we did not conduct the release study to explain the phenomenon. We think this effect was due to the higher concentration of PRE or high doses of pentacyclic triterpenes could decrease cell viability as same as the results from the previous study [34].

Comment 10

             Lines 456-470: The data is presented in Table 6. Discuss only the most relevant results.

Response #10

            Thank you for your suggestion. We rewrote sentences in the revised version, as shown below.

Original version

It was found that the enhancement effect on HDF cell proliferation was achieved by the MTT method. The results showed that microemulsions containing 1% and 2.5% PRE exhibited a potent proliferative effect (123.91-136.30%) at 3 and 10 μg/mL. While microemulsion containing 5% PRE acted as a proliferative inducer (108.54 ± 4.67%) at low concentrations (1 μg/mL), and microemulsion base showed the lowest percent proliferation (<100%) on HDF cells. The microemulsion containing 1% PRE showed the highest proliferation property at concentrations 10 μg/mL (136.30 ± 3.93%), 2.5% PRE showed the highest proliferation property at concentrations 3 μg/mL (123.91 ± 3.96%), and 5% PRE showed the highest proliferation property at concentrations 1 μg/mL (108.54 ± 4.67%). Each sample acted as a low proliferative at higher concentrations. Our results suggested that pentacyclic triterpenes content in each formulation plays an important role in cell proliferation. It is consistent with previous test results on cell proliferation of PRE that showed the best activity at the optimum concentration. The increasing cell viability of each sample at 0.3, 1, 3, 10, and 30 μg/mL are presented in Table 6. The experiment supported that the microemulsions containing PRE stimulate the growth of HDF cells.

Revised version

It was found that the enhancement effect on HDF cell proliferation was achieved by the MTT method. The results showed that microemulsions containing 1% and 2.5% PRE exhibited a potent proliferative effect (123.91-136.30%) at 3 and 10 μg/mL. While microemulsion containing 5% PRE acted as a proliferative inducer (108.54 ± 4.67%) at low concentrations (1 μg/mL), and microemulsion base showed the lowest percent proliferation (<100%) on HDF cells. The microemulsion containing 1%, 2.5%, and 5% PRE showed the highest proliferation property at concentrations of 10, 3, and 1 μg/mL, respectively. All samples acted as a low proliferative at higher concentrations. Our results suggested that pentacyclic triterpenes content in each formulation plays an important role in cell proliferation. It is consistent with previous test results on cell proliferation of PRE that showed the best activity at the optimum concentration [43, 44]. The increasing cell viability of each sample at 0.3, 1, 3, 10, and 30 μg/mL are presented in Table 6. It should be noted that all microemulsions containing PRE stimulate the growth of HDF cells.

Comment 11

             Lines 616-624: Do the authors consider these differences in collagen content relevant? Compare with other emulsions with the same application and discuss it.

Response #11

Thank you for your suggestion. We rewrote sentences in the revised version, as shown below.

Original version

Numerous studies have documented the action of triterpenes from C. asiatica, such as MS, MA, and AA, on the synthesis of collagen related to the wound-healing process [37]. The experiments of the Sircol collagen assay provide the information that the substances can be added to the microemulsion formula to stimulate collagen production. The collagen content in the medium of fibroblasts incubated with the test sample was measured. Collagen production on HDF and HGF cells by microemulsion base and microemulsion containing PRE at various concentrations were evaluated and compared with the control. For HDF cells, it was found that the microemulsion containing 1% PRE exhibited the highest collagen content of 55.80 ± 0.54 μg/mL followed by base (53.80 ± 1.15 μg/mL), 2.5% PRE (53.10 ± 1.91 μg/mL) and 5% PRE (51.50 ± 0.94), respectively. For HGF cells, it was found that the collagen content of microemulsion containing 1% PRE also showed the highest collagen content at 52.80 ± 0.54 μg/mL followed by 5% PRE (50.10 ± 0.94), 2.5% PRE (49.60 ± 1.91) and base (46.80 ± 0.54), respectively. From the result mentioned above, after the statistical test, it could be concluded that all formulations tested revealed slightly significant differences better than the control (p<0.05). However, all formulations tested did not show any significant difference between each formulation. The results obtained are shown in Table 10:

Revised version

Numerous studies have documented the action of triterpenes from C. asiatica, such as MS, MA, and AA, on the synthesis of collagen related to the wound-healing process [46,47,48]. The experiments of the Sircol collagen assay provide the information that the substances can be added to the microemulsion formula to stimulate collagen production. The collagen content in the medium of fibroblasts incubated with the test sample was measured. Collagen production on HDF and HGF cells by microemulsion base and microemulsion containing PRE at various concentrations were evaluated and compared with the control. For HDF cells, it was found that the microemulsion containing 1% PRE exhibited the highest collagen content of 55.80 ± 0.54 μg/mL followed by base (53.80 ± 1.15 μg/mL), 2.5% PRE (53.10 ± 1.91 μg/mL) and 5% PRE (51.50 ± 0.94), respectively. For HGF cells, it was found that the collagen content of microemulsion containing 1% PRE also showed the highest collagen content at 52.80 ± 0.54 μg/mL followed by 5% PRE (50.10 ± 0.94 μg/mL), 2.5% PRE (49.60 ± 1.91 μg/mL) and base (46.80 ± 0.54 μg/mL), respectively. Combining with statistical analysis, it could be concluded that all formulations tested revealed slightly significant differences better than the control (p<0.05). However, all formulations tested did not show a significant difference between each formulation. Collagen production in both cells was not different between base and various concentrations. This may be due to the coconut oil in the preparation, which could also promote collagen production [49,50]. The results revealed that the combination of PRE and coconut oil facilitated collagen production in fibroblast cells. The results obtained are shown in Table 10.
